# Identification of phenotypically, functionally, and anatomically distinct stromal niche populations in human bone marrow based on single-cell RNA sequencing

Hongzhe Li[1]*, Sandro Bräunig[1], Parashar Dhapolar[1], Göran Karlsson[1], Stefan Lang[1], Stefan Scheding[1,2]*

[1]Division of Molecular Hematology and Stem Cell Center, Lund University, Lund, Sweden; [2]Department of Hematology, Skåne University Hospital, Lund, Sweden

**Abstract** Hematopoiesis is regulated by the bone marrow (BM) stroma. However, cellular identities and functions of the different BM stromal elements in humans remain poorly defined. Based on single-cell RNA sequencing (scRNAseq), we systematically characterized the human non-hematopoietic BM stromal compartment and we investigated stromal cell regulation principles based on the RNA velocity analysis using scVelo and studied the interactions between the human BM stromal cells and hematopoietic cells based on ligand-receptor (LR) expression using Cell-PhoneDB. scRNAseq led to the identification of six transcriptionally and functionally distinct stromal cell populations. Stromal cell differentiation hierarchy was recapitulated based on RNA velocity analysis and in vitro proliferation capacities and differentiation potentials. Potential key factors that might govern the transition from stem and progenitor cells to fate-committed cells were identified. In situ localization analysis demonstrated that different stromal cells were localized in different niches in the bone marrow. In silico cell-cell communication analysis further predicted that different stromal cell types might regulate hematopoiesis through distinct mechanisms. These findings provide the basis for a comprehensive understanding of the cellular complexity of the human BM microenvironment and the intricate stroma-hematopoiesis crosstalk mechanisms, thus refining our current view on human hematopoietic niche organization.

## Editor's evaluation

The manuscript by Li and coworkers is a landmark characterization of sorted human non-hematopoietic bone marrow cells by scRNA-seq, which predicts their potential lineage relationships and possible interactions with mature and immature hematopoietic cells. Transcriptionally-different stromal cell subsets are identified convincingly, and their lineage relationships, cell-cell interactions and possible specialized functions are predicted from in-silico studies, paving the way for future necessary functional validation studies. This resource significantly adds to the current understanding of human non-hematopoietic bone marrow stromal cells and their hematopoietic regulatory functions.

*For correspondence:
hongzhe.li@med.lu.se (HL);
stefan.scheding@med.lu.se (SS)

**Competing interest:** The authors declare that no competing interests exist.

## Introduction

Bone marrow (BM) is the principal site of hematopoiesis in adult humans. In the BM, hematopoietic stem cells (HSCs) and their progenies are contained in specialized microenvironments which regulate HSC maintenance and differentiation.

Our understanding of the BM hematopoietic microenvironment (HME) has evolved considerably over the past decade through a number of landmark studies that have identified the cellular identity, anatomy, and functions of different murine HME components (*Baryawno et al., 2019*; *Tikhonova et al., 2019*; *Wolock et al., 2019*; *Baccin et al., 2020*). Skeletal and stromal cell populations, endothelial cells and other non-hematopoietic cells but also differentiated hematopoietic cells have been described as important HME elements in mice (*Méndez-Ferrer et al., 2008*; *Urbieta et al., 2010*; *Yamazaki et al., 2011*; *Ding et al., 2012*; *Casanova-Acebes et al., 2013*; *Zhao et al., 2014*; *Chan et al., 2018*).

In contrast, important HME elements in human BM remain poorly defined, which is due to the fact that the identity and function of the stromal stem/progenitor cells have been difficult to investigate (*Li et al., 2016*). Nevertheless, a number of recent studies have provided the first important insights into the complexity of the human BM and the potential diverse functional roles of BM stromal cells (*Chan et al., 2018*; *de Jong et al., 2021*; *Triana et al., 2021*; *Wang et al., 2021*).

Based on single-cell RNA sequencing (scRNAseq) technology, we herein investigated the human non-hematopoietic BM cell compartment aiming to resolve the composition of the human HME at the highest possible resolution, to identify potential novel marrow stromal subsets and cellular hierarchies as well as to establish functional relationships between stromal and hematopoietic elements.

## Materials and methods

### Human bone marrow mononuclear cell isolation

Human bone marrow (BM) cells were collected at the Hematology Department, Skåne University Hospital Lund, Sweden, from consenting healthy donors by aspiration from the iliac crest as described previously (*Li et al., 2014*). Bone marrow mononuclear cells (BM-MNC) were isolated by density gradient centrifugation (LSM 1077 Lymphocyte, PAA, Pasching, Austria). The plasma layer supernatant containing adipocytes was discarded. The use of human samples was approved by the Regional Ethics Review Board in Lund, Sweden.

### Flow cytometry and fluorescence activated cell sorting (FACS)

For sorting of BM-MNCs for scRNAseq analysis, freshly isolated BM-MNCs were incubated in blocking buffer [DPBS w/o Ca2+, Mg2+, 3.3 mg/ml human normal immunoglobulin (Gammanorm, Octapharm, Stockholm, Sweden), 1% FBS (Invitrogen)], followed by staining with monoclonal antibodies against CD271, CD235a, and CD45. Sorting gates were set according to the corresponding fluorescence-minus-one (FMO) controls and cells were sorted on a FACS Aria II or Aria III (BD Bioscience, Erembodegem, Belgium). Dead cells were excluded by 7-Amino-actinomycin (7-AAD, Sigma) staining and doublets were excluded by gating on FSC-H versus FSC-W and SSC-H versus SSC-W. Cells were directly sorted into PBS with 0.04% BSA for scRNAseq analysis. Sorting of BM-MNCs for CFU-F or stromal culture was performed using DAPI for dead cell exclusion and cells were directly sorted into stromal cell culture medium [StemMACS MSC Expansion Media, human (Miltenyi Biotec, Bergisch Gladbach, Germany)].

### CFU-F (colony-forming unit, fibroblast) assay and generation of cultured stromal cells (cSCs)

Unsorted and FACS-sorted primary BM-MNC were cultured at plating densities of $5 \times 10^4$ cells/cm$^2$ cells and 10–50 cells/cm$^2$, respectively. Colonies (≥40 cells) were counted after 14 days (1% Crystal Violet, Sigma). Assays were set up in duplicates or triplicates. For single-cell CFU-F assays, cells were sorted into 96-well plates, cultured in stromal cell culture medium, and colonies were counted after 3 weeks. Thereafter, cells were harvested and culture-expanded for use in subsequent experiments. Medium was changed weekly and cultured stromal cells were passaged at 80% confluency after trypsinization (0.05% trypsin/EDTA, Invitrogen, Carlsbad, USA).

## In vitro differentiation assays

Cultured mesenchymal stromal cells (cMSCs) were differentiated towards the adipogenic, osteoblastic, and chondrogenic lineage as described previously (*Li et al., 2014*). Briefly, cells were cultured for 14 days in AdipoDiff medium (Miltenyi) and cells were stained with Oil Red O (Sigma). For osteogenic differentiation, cells were cultured in osteogenesis induction medium for 21 days and calcium depositions were detected by Alizarin Red staining (Sigma). Osteogenesis induction medium: standard MSC medium supplemented with 0.05 mM L-ascorbic-acid-2-phosphate (Wako Chemicals, Neuss, Germany), 0.1 µM dexamethasone and 10 mM β-glycerophosphate (both from Sigma).

Chondrogenic differentiation was induced by culturing cell pellets ($2.5 \times 10^5$ cells/pellet) for 28 days in chondrogenesis induction medium. Chondrogenesis induction medium: DMEM-high glucose supplemented with 0.1 µM dexamethasone, 1 mM sodium pyruvate, 0.35 mM L-proline (all from Sigma), 0.17 mM ascorbic acid, 1% ITS +culture supplements (BD Biosciences) and 0.01 µg/ml TGF-ß3 (R&D Systems). Pellets were paraformaldehyde (PFA)-fixed and frozen in O.C.T. Compound (Sakura, Zoeterwoude, Netherlands). Cryosections were stained with Alcian Blue, and nuclei were stained with Nuclear Fast Red (both from Sigma). Sections were analyzed with a Nikon Eclipse TS100 microscope equipped with a Nikon DS-L3 digital camera.

## Antibodies

The following antibodies were used for FACS analysis and cell sorting: CD235a-PE-Cy5 [clone GA-R2 (HIR2)], CD71- PE-Cy5 (clone M-A712), CD45-APC-Cy7 (clone 2D1), CD52-Alexa Fluor (AF) 488 (clone 4C8), CD56 (NCAM1)- Brilliant Violet 605 (clone Leu-19, NKH1), CD81-PE (JS-81) (all BD Bioscience), LEPR-PE (clone 52263, R&D Systems) CD271-APC (clone ME20.4–1 .H4, Miltenyi).

## Immunofluorescence staining

Human bone marrow paraffin sections (5 µm) were deparaffinized and rehydrated following standard protocols. Following antigen retrieval with antigen retrieval solution (DAKO, S1699) and blocking with goat serum (Invitrogen, #31873), slides were stained sequentially scanned, and restained following fluorescence bleaching of primary coupled antibodies in PBS with 50 mM $Na_2CO_3$ and 3% $H_2O_2$. An OlympusVS120 slide scanner was used for scanning. Antibodies used were CD45-AlexaFlour 647 (Bio-Rad, MCA87A647T), CD56 (NCAM1)-AF647 (Biolegend, 318313), mouse anti-human CD271 (R&D Systems, MAB367), and rabbit anti-human CD81 (Novus Biologicals, NBP1-77039) with secondary goat anti-mouse AF647 (Jackson ImmunoResearch, 115-605-166) and goat anti-rabbit AF488 (Jackson ImmunoResearch, 111-545-003) with DAPI (Sigma, D956410MG) as a nuclear counterstain, respectively.

For SPP1 staining, after deparaffinization, rehydration, and antigen retrieval, slides were permeabilized with 0.1% Triton X100 (Sigma Ultra, T9284) solution at 37 °C for 15 min, blocked using donkey serum (Jackson Immuno Research, 017-000-121), and stained overnight at 4 °C with goat anti-human SPP1 antibody (R&D Systems, AF1433-SP), followed by secondary staining with donkey anti-goat AF594 (Jackson ImmunoResearch, 706-585-147) and staining with CD56 (NCAM1)-AF647 plus DAPI. Slides were then scanned, bleached, and incubated with rabbit anti-human CD45 (Sigma-Aldrich, HPA000440) and mouse anti-human CD271 overnight, followed by secondary antibody staining with goat anti-mouse AF647 and goat anti-rabbit AF488 and scanned again.

For confocal laser scanning, slides went through the same standard deparaffinization, rehydration, and antigen retrieval procedure. For all primary antibody stainings, blocking was done with normal goat serum, followed by overnight incubation at 4 °C with rabbit and mouse primary antibodies. Thereafter, slides were incubated with secondary goat anti-mouse AF647 and goat anti-rabbit AF488 for 1 hr at room temperature (RT). Primary antibodies used were mouse anti-human CD271 (R&D Systems, MAB367), rabbit anti-human CD81 (Novus Biologicals, NBP1-77039), mouse anti-human CD81 (Abcam, ab79559), rabbit anti-human CD45 (Sigma-Aldrich, HPA000440), rabbit anti-human CD56 (Sigma-Aldrich, 156 R-96), and mouse anti-human CD56 (Fisher scientific, 10027562). For SPP1, NCAM1, CD271 co-stainings, slides were first blocked with goat serum and incubated with rabbit anti-human NCAM1 and mouse anti-human CD271 antibodies overnight at 4 °C. Thereafter, slides were incubated with secondary goat anti-mouse AF647 and goat anti-rabbit AF488 for 1 hr at RT followed by permeabilization with 0.1% Triton X100 (Sigma Ultra, T9284) solution at 37 °C for 15 min. After blocking with donkey serum donkey anti-human SPP1 antibody was incubated overnight at 4 °C

followed by donkey anti-goat AF594 (Jackson ImmunoResearch, 706-585-147) incubation for 1 hr at RT.

After coverslip mounting, all slides were scanned with a Zeiss 780 Confocal Laser Scanning Microscope. 3D orthographic cross-section views and intensity profiles were generated with the ZEN Microscopy Software (Zeiss).

## Immunofluorescence image processing

All image data were analyzed and processed using arivis Vision4D. The sequential regions were aligned and overlayed based on nuclei positions. Sequentially used fluorophores were assigned pseudo colors. The obtained five color stacks were cropped to an overlapping region and a maximum intensity projection of the three-layer z-stack was generated.

## Co-culture of BM CD34$^+$ cells with stromal cells

FACS-sorted stromal cells (OC: CD45$^{low/-}$CD235a$^-$CD71$^-$CD271$^+$CD52$^-$NCAM1$^+$; MSSC: CD45$^{low/-}$CD235a$^-$CD71$^-$CD271$^+$CD52$^-$NCAM1$^-$CD81$^{++}$) were culture expanded and plated as adherent feeder cells into 96-well plates at 10,000 cells per well and cultured in StemMACS MSC Expansion medium for 1 day. Then, medium was removed and 5000 BM CD34$^+$ cells were added in serum-free expansion medium (SFEM, STEMCELL Technologies) supplemented with or without stem cell factor, thrombopoietin, and FLT-3 ligand (all at 25 ng/ml). Expanded cells were harvested, counted and the expression of CD34 and CD90 was analyzed by flow cytometry after 7 days of co-culture.

## Single-cell RNA sequencing and data analysis

### Single-cell RNA sequencing

Single-cell RNA sequencing (scRNAseq) was performed by the Single-Cell Genomics Platform at the Center for Translational Genomics at Lund University. FACS-sorted BM-MNC populations analyzed were CD45$^{low/-}$CD235a$^-$ cells from four donors (two males, two females, ages 19, 22, 52, and 53 years) and CD45$^{low/-}$CD235a$^-$CD271$^+$ from five donors (three males, two females, ages 21, 25, 32, 58, and 61 years) (*Supplementary file 1*). Donors younger than 35 years or older than 50 were considered as young donors and old donors, respectively. FACS-sorted BM-MNC cells were encapsulated into emulsion droplets as single cells using the Chromium Controller (10 X Genomics). scRNAseq libraries were constructed using the Chromium Single Cell Gene Expression 3' v3 Reagent Kit according to the manufacturer's instructions. Reverse transcription and library preparation was performed on a C1000 Touch Thermal Cycler (Bio-Rad). Amplified cDNA and final libraries concentrations were measured with a Qubit 4 Fluorometer (Invitrogen) using the dsDNA High Sensitivity Assay (Invitrogen). cDNA and library traces were evaluated on a TapeStation (Agilent) using High Sensitivity D5000 and D1000 Screen Tapes (Agilent). Individual libraries were diluted to 1.5 nM and pooled for sequencing. Pools were sequenced on a NovaSeq 6000 (Illumina) aiming for a sequencing depth of 50,000 reads per cell.

## Pre-processing of scRNA-seq data, cell filtering, and normalization

scRNAseq data of both CD45$^{low/-}$CD235a$^-$ and CD45$^{low/-}$CD235a$^-$CD271$^+$ populations were processed and combined for further data analysis. Sequencing data were demultiplexed and UMI-collapsed using the CellRanger toolkit (version 3.1, 10 X Genomics). Common QC metrics including cell filtration, downsampling, gene filtration, mitochondria and ribosomal reads exclusion were used to identify low-quality cells. Reads were mapped against the human GRCh38 genome paired with the gencode. v31 gene level data using the kallisto/bustools package (version 0.46.0 resp. 0.39.3). Resulting count metrics from the kallisto/bustools were analyzed using Python scanpy/velocyto. Mitochondrial and ribosomal reads were excluded and cells with less than 1000 (diploid) detected UMIs were removed from the analysis. Expression was normalized using the scanpy.pp.downsample_counts function to 1000 UMIs. Finally, genes not expressed in at least 10 cells were also removed from the further analysis.

### Dimensionality reduction

Dimensionality reduction was performed using gene expression data for a subset of 3000 variable genes with the scanpy.pp.highly_variable_genes function. UMAP was generated with the default setting of Scanpy version 1.6.0 using the top 50 principal components (PC).

## Clustering and visualization

Graph-based clustering of the PCA-reduced data was performed using the Louvain method implemented in scanpy. The resolution parameter was set to 11, which resulted in 103 preliminary clusters that were merged based on the correlation of mean expression for each cluster. Clusters with correlations greater than 0.95 were merged, resulting in the final clusters displayed in *Figure 1B*. Cell type annotation was performed based on the published marker genes expression (*Supplementary file 2*; *Baryawno et al., 2019*).

## Differential expression of gene signature

For each cluster, we used the scanpy.tl.rank_genes_groups function with default values to identify genes that had a significant fold change of higher than 2 in comparison to the other clusters.

## RNA velocity analysis

Bone marrow stromal cell dynamics were analyzed using the scVelo package (version 0.2.2) in Scanpy 1.6.0 (*Svensson and Pachter, 2018*; *Bergen et al., 2020*). The data were processed using default parameters following preprocessing as described in scVelo package. Analysis of cellular trajectory inference by RNA velocity was performed using dynamical modeling, which is a generalization of the original RNA velocity method and which allows for characterization of multiple transcriptional states. Latent time analysis, pseudotime analysis and velocity confidence were computed using default parameters. Extrapolated states were then projected on the UMAP embedding produced during the initial analysis step. The clustering and visualization were repeated using the same parameters as above for the selected stromal clusters forming the continuum.

For velocity analysis, the samples were pre-processed using functions for detection of the minimum number of counts, filtering and normalization using scv.pp.filter_and_normalise and followed by scv.pp.moments function. The gene-specific velocities were then calculated using scv.tl.velocity with mode set to stochastic, and visualized using scv.pl.velocity_embedding function. In addition, we used scv.tl.latent_time function to infer a shared latent time from splicing dynamics and plotted the genes along a time axis sorted by expression along dynamics using scv.pl.heatmap function. Pseudotime trajectory and velocity confidence were computed using velocity_pseudotime, and scv.tl.velocity_confidence functions.

## Cell-cell communication analysis

Cell-cell interaction analysis was conducted with CellPhoneDB2 (*Efremova et al., 2020*), which is a Python-based analytical tool for calculating the interaction between ligands and receptors between different cell populations. The mean and p value of each ligand-receptor interaction between different clusters were calculated by CellPhoneDB2. To illustrate the strength of specific pathways between different clusters, several common pathways with high mean values were selected for visualization. Dot plots were generated with R studio (Version 1.3.1093). The list of ligand-receptor interaction pairs is shown in *Supplementary file 6*.

## Code availability

The code generated to analyze the single-cell datasets in this study is available through GitHub at https://github.com/Hongzhe2022/MSC_BM_scripts, copy archived at *Stefan, 2022*.

## Results

### Single-cell RNAseq identifies distinct cell populations in the human bone marrow microenvironment

We explored the cellular composition of the normal BM stroma by scRNAseq profiling of the non-hematopoietic cell-containing CD45$^{low/-}$CD235a$^-$ population from healthy donors (*Figure 1A*). The well-known low frequency of the BM stromal stem/progenitor cells (around 0.001–0.01% of BM cells) (*Pittenger et al., 1999*) imposed challenges to detect CXCL12-expressing stromal cells in the CD45$^{low/-}$CD235a$^-$ population (*Figure 1—figure supplement 1A*). In order to also capture and be able to analyze extremely rare stromal cell subsets, we therefore also sorted CD45$^{low/-}$CD235a$^-$CD271$^+$ cells,

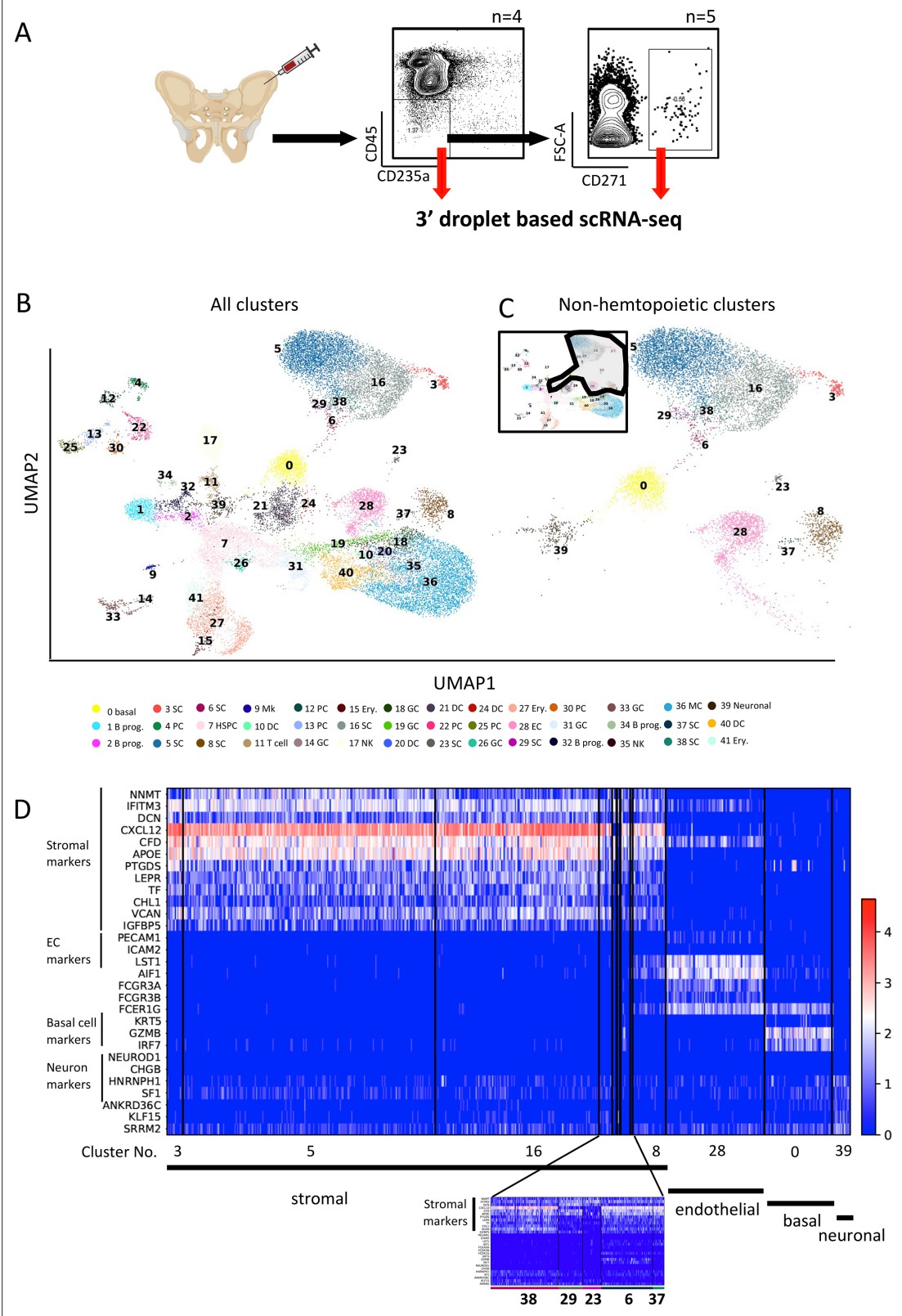

**Figure 1.** Single-cell atlas of human bone marrow CD45$^{low/-}$CD235a$^-$ cells. (**A**) Overview of the study design including gating strategies for isolation of human bone marrow CD45$^{low/-}$CD235a$^-$ and CD45$^{low/-}$CD235a$^-$CD271$^+$ cells. (**B**) Uniform Manifold Approximation and Projection (UMAP) display of single-cell transcriptomic data of human bone marrow CD45$^{low/-}$CD235a$^-$ cells containing enriched CD45$^{low/-}$CD235a$^-$CD271$^+$ cells to allow for a detailed analysis of rare stromal cell subpopulations. Data are shown for a total of n=25067 cells [5704 CD45$^{low/-}$CD235a$^-$ cells (22.76%) and 19363 CD45$^{low/-}$

*Figure 1 continued on next page*

*Figure 1 continued*

CD235a⁻CD271⁺ cells (77.24%)] from a total of nine healthy donors. Color legend indicates cluster numbers and annotations. Basal, basal cell-like cluster; B prog., B cell progenitors; SC, stromal cells; PC, plasma cells; HSPC, hematopoietic stem and progenitor cells; Mk, megakaryocytes; DC, dendritic cells; GC, granulocytes; Ery., erythroid cells; NK, natural killer cells; EC, endothelial cells; Neuronal, neuronal cell-containing cluster. (**C**) UMAP display of non-hematopoietic clusters (n=9686 cells). The circled area in the overview (top left corner) indicates the non-hematopoietic clusters selected from (**B**). (**D**) Heatmap of representative differentially expressed genes for each of the non-hematopoietic clusters in (**C**). For stromal marker identification, the top 100 significant differentially expressed genes from each stromal cluster were selected to identify overlapping genes. Three genes shared by all nine clusters were identified in this comparison (NNMT, IFITM3, and DCN). Nine more genes were identified when OC clusters 23 and 29 (identified as OCs in *Figure 2A*) were removed from the comparison, as OCs showed considerably different gene expression profiles as compared with other stromal clusters. Cluster numbers and corresponding cell types are indicated under the heatmap. The scale bar indicates gene expression levels. EC, endothelial cells. A blow-up heatmap for clusters 38, 29, 23, 6, and 37 is shown under the main heatmap for better visualization.

The online version of this article includes the following figure supplement(s) for figure 1:

**Figure supplement 1.** Cluster annotation of human bone marrow CD45low/-CD235a cells.

which are highly enriched for CXCL12-expressing BM stromal stem and progenitor cells (*Figure 1A*, *Figure 1—figure supplement 1B*; *Tormin et al., 2011*; *Li et al., 2014*). As CD45^low/-^CD235a⁻CD271⁺ cells represent a subset of CD45^low/-^CD235a⁻ cells, we combined the two datasets which allowed for a detailed analysis of the structural and developmental organization of the BM stroma at the highest possible resolution.

In total, our dataset was comprised of 25,067 cells (median of 856 genes and 2937 UMIs per cell, *Figure 1—figure supplement 1C*) which formed 42 clusters corresponding to distinct cell types and differentiation stages, respectively (*Figure 1B*, *Figure 1—figure supplement 1D–F*, *Supplementary file 1*; *Supplementary file 2*; *Supplementary file 3*). As expected (*Boulais et al., 2018*; *Baryawno et al., 2019*), scRNAseq analysis of CD45^low/-^CD235a⁻ cells captured several hematopoietic cell types including a cluster of hematopoietic stem/progenitor cells (HSPCs) enriched for CD34 expression and other HSPC markers (PROM1, CRHBP, AVP, MLLT3, FAM30A, GATA1) (*Figure 1—figure supplement 1G*, and *Supplementary file 2* and *Supplementary file 4*). Importantly, non-hematopoietic cells including stromal cells as well as non-stromal cells, such as endothelial cells (cluster 28), KRT5-enriched basal cell-like cluster (cluster 0), and a cluster of cells enriched for neuronal markers expression (cluster 39) could be clearly identified (*Figure 1C and D* and *Figure 1—figure supplement 1H*). These clusters were detected in both CD45^low/-^CD235a⁻ and CD45^low/-^CD235a⁻CD271⁺ cells (*Figure 1—figure supplement 1D–E*). Of note, following the exclusion of hematopoietic cells and the other non-stromal cell types, we identified nine stromal clusters (clusters 3, 5, 6, 8, 16, 23, 29, 37, 38, *Figure 1B and C*), which only became detectable after CD45^low/-^CD235a⁻CD271⁺ cells were included in the analysis. These findings thus validated our CD271⁺ cell enrichment strategy and proved that a high-resolution transcriptomic analysis of the thus far poorly defined human BM stromal compartment would not have been achieved without CD45^low/-^CD235a⁻CD271⁺ enrichment.

## Cellular heterogeneity of stromal progenitors

The nine stromal progenitor populations shared a unique gene expression profile, including both well-established stromal markers as well as novel markers (*Supplementary file 2*; *Supplementary file 4*; *Supplementary file 7*; *Li et al., 2014*). These markers clearly distinguished stromal cells from the other non-hematopoietic cell types as demonstrated by the differential expression of several stromal genes (*Figure 1D*, and *Figure 1—figure supplement 1H*).

The stromal clusters were then annotated based on their expression of previously reported BM stromal markers (*Churchman et al., 2012*; *Li et al., 2014*), which allowed us to discriminate six different cell types, that is multipotent stromal stem cells (MSSCs), highly adipocytic gene-expressing progenitors (HAGEPs), balanced progenitors, pre-osteoblasts, osteochondrogenic progenitors (OCs), and pre-fibroblasts (*Figure 2A–B*, *Figure 2—figure supplement 1A–B*).

Cluster 3 was annotated as MSSCs with multi-lineage differentiation capacity based on its high expression of stromal markers (CXCL12, LEPR, DCN, PTGDS) and genes indicating both adipogenic (CEBPD, LPL, PLIN1, ADIPOQ, CCL2, PPARG) and osteogenic (GAS6, FBN1, ALPL, RUNX1, SPARCL1, CDH11) differentiation capacities (*Figure 2C and D*). The annotation of cluster 3 as MSSCs was further confirmed by velocity analysis as shown in *Figure 3A*.

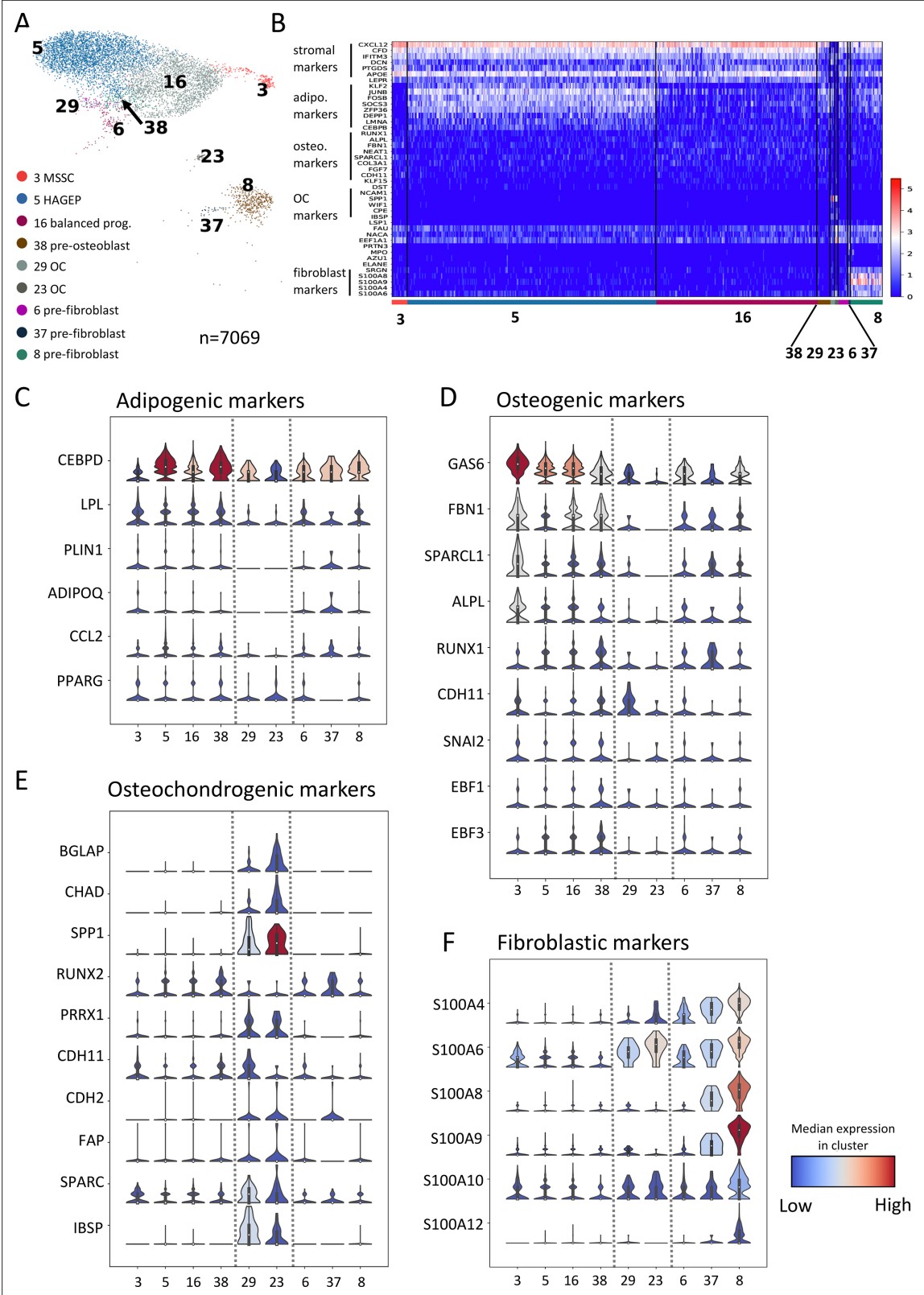

**Figure 2.** scRNAseq reveals distinct gene expression patterns in different bone marrow stromal cell populations. (**A**) UMAP display of the nine stromal subsets (n=7069 cells). Color legends indicate stromal cluster numbers and annotations. MSSC, multipotent stromal stem cells; OC, osteochondrogenic progenitors. (**B**) Single cell heatmap of representative differentially expressed genes in each cluster shown in (**A**). A blow-up heatmap for clusters 38, 29, 23, 6, and 37 is shown in *Figure 2—figure supplement 1A* for better visualization. The scale bar indicates gene expression levels. (**C–F**) Stacked violin

*Figure 2 continued on next page*

Figure 2 continued

plots of adipogenic- (**C**), osteogenic- (**D**), osteochondrogenic- (**E**), and fibroblastic (**F**) markers in different stromal clusters. Dashed lines separate MSSCs, HAGEPs, balanced progenitors, and pre-osteoblast (left; cluster 3, 5, 16, and 38), OCs (middle; cluster 29, 23), and pre-fibroblasts (right; cluster 6, 37, 8). The boxes in the violin plots indicate the lower, median and upper quartiles.

The online version of this article includes the following figure supplement(s) for figure 2:

**Figure supplement 1.** Differential expression of markers in stromal clusters.

Cells in cluster 5 expressed adipogenic differentiation markers as well as a group of stress-related transcription factors such as FOS, FOSB, JUNB and EGR1, some of which have been shown to mark bone marrow adipogenic progenitors (*Figure 2B–C* and *Figure 2—figure supplement 1C*; *Ambrosi et al., 2017*). Osteogenic differentiation markers were also expressed but at considerably lower levels compared to adipogenic genes (*Figure 2D*). Thus, cluster 5 was termed as highly adipocytic gene-expressing progenitors (HAGEPs). Cluster 16 on the other hand, showed similar expression levels of adipogenic and osteogenic markers and was therefore annotated as balanced progenitors (*Figure 2A–D*). We observed a gradual increase of more mature osteogenic markers such as RUNX1, CDH11, EBF1, and EBF3 from cluster 5 to cluster 16 and to cluster 38 (*Figure 2D*). Accordingly, cluster 38 was denoted as pre-osteoblasts. Clusters 29 and 23 were characterized by prominent expression of osteochondrogenic markers including BGLAP, CHAD, SPP1(OPN), RUNX2, CDH11, CDH2, IBSP (BSP), and SPARC (*Figure 2B and E*, *Figure 2—figure supplement 1A*). Hence, these two clusters were assigned as osteochondrogenic progenitors (OCs). While both adipogenic and osteogenic marker expression was detected in clusters 6, 37 and 8, they also expressed several hematopoiesis-related markers such as SRGN, CD52, CD37, CD48, and PTPRC (*Figure 2B* and *Figure 2—figure supplement 1A* and D). Furthermore, S100 genes, which encode a group of calcium-binding cytosolic proteins, were expressed almost exclusively in cluster 8 (*Figure 2B and F*), and the previously reported murine fibroblast markers, S100A4 and S100 A6 (*Słomnicki and Leśniak, 2010*, *Baryawno et al., 2019*) were highly expressed in all three clusters. Based on this expression pattern, clusters 6, 37, and 8 were subsequently annotated as pre-fibroblasts.

## Inferred trajectories of stromal cell gene expression reconstruct the temporal sequence of transcriptomic events during stromal cell differentiation

To resolve the differentiation dynamics of stromal subsets and infer the directionality of individual cells during differentiation, RNA velocity analysis was performed using the scVelo toolkit based on the changes in the spliced/unspliced ratio of mRNA counts (*Bergen et al., 2020*). *Figure 3A and A'* illustrate the transcriptional dynamics as indicated by streamlines with arrows pointing from the MSSC cluster to different progenitor clusters. RNA velocity analysis predicted that the MSSC cluster has two main developmental directions (*Figure 3A'*) with the first originating from MSSCs (cluster 3) and moving towards HAGEPs (cluster 5), followed by OCs (cluster 29) and terminating at the pre-osteoblast cluster (cluster 38). The second main differentiation path, which also originated from MSSCs (cluster 3), was directed towards balanced progenitors and ended at either the pre-osteoblasts or the pre-fibroblasts stage. A small fraction within HAGEPs also showed development potential towards balanced progenitors and pre-osteoblasts (*Figure 3A and A'*). Latent time analysis, which represents the cell's internal clock and approximates the real time experienced by cells as they differentiate, also identified MSSCs as the root cell population and placed pre-osteoblasts close to the terminal state of differentiation (*Figure 3B*).

To provide a more precise approximation of the main differentiation stages, we performed latent time and pseudotime analysis on only those clusters that formed a continuum of cellular differentiation states in UMAP, that is MSSCs, HAGEPs, balanced progenitors, OCs (cluster 29), pre-osteoblasts and one of the pre-fibroblast clusters (cluster 6) (*Figure 3C*). Thereby, disturbances of the data analysis by larger gaps in cellular state transitions between the clusters within the continuum and the outlying OCs (cluster 23) and pre-fibroblasts (clusters 8 and 37) were avoided (*Figure 3A*). The results of these analyses using two different methods which are based on different mathematical algorithms further corroborated our initial conclusion that the MSSC cluster was at the apex of a differentiation hierarchy with downstream differentiation paths into HAGEPs, balanced progenitors, OCs and pre-osteoblasts

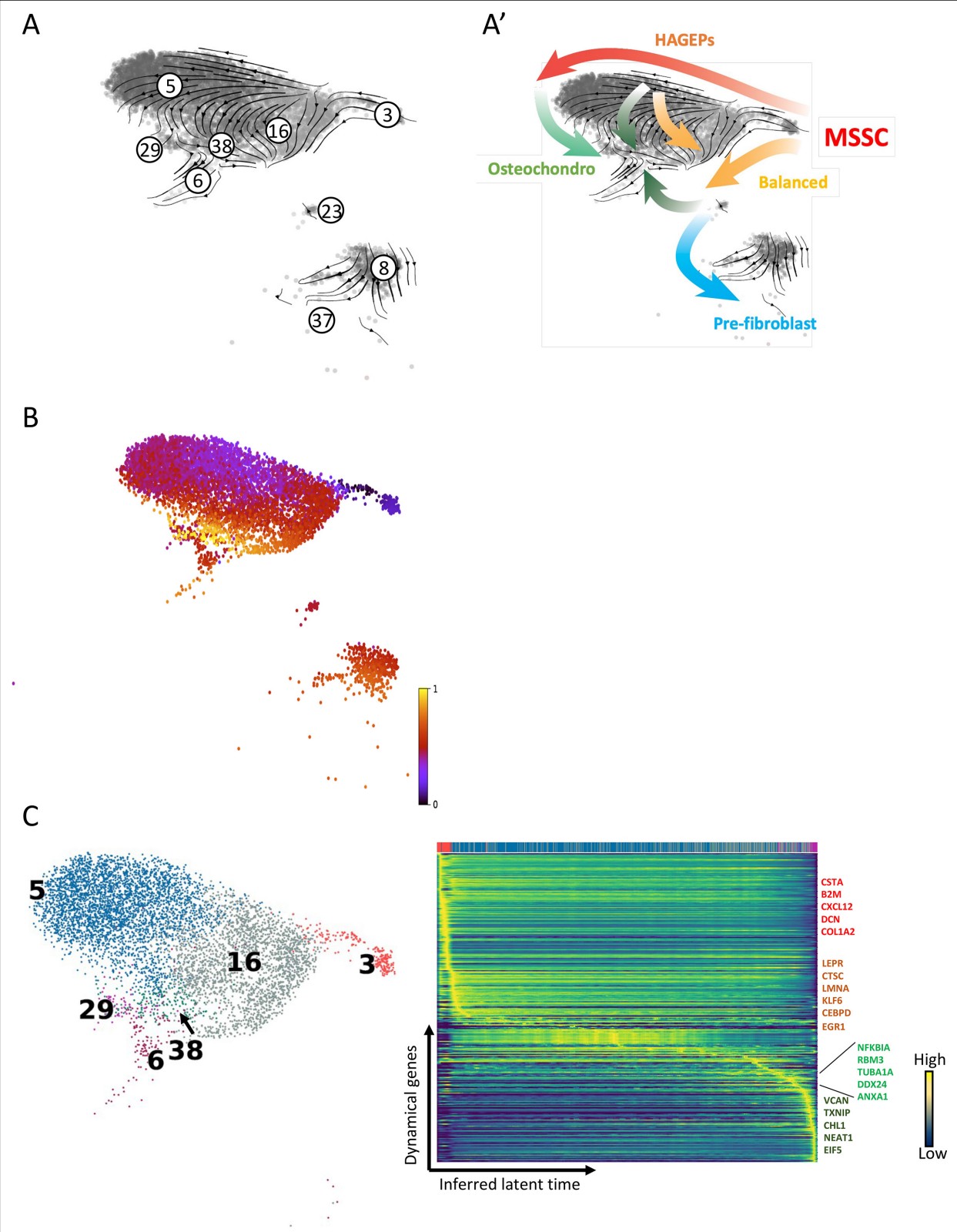

**Figure 3.** RNA velocity analysis reconstructs the temporal sequence of transcriptomic events of stromal cells. (**A and A'**) Single cell velocities of the nine stromal clusters visualized as streamlines in a UMAP. Black arrows indicate direction and thickness indicates speed along the stromal cell development trajectory. (**A'**) Colored thick arrows indicate the main directions of stromal cell developmental paths. Colors correspond with the differentiation destination. Red: HAGEPs; yellow: balanced progenitors; light green: OCs; dark green: pre-osteoblasts; blue: pre-fibroblasts. (**B**) UMAP display of

*Figure 3 continued on next page*

*Figure 3 continued*

stromal cells colored by inferred latent time. Inferred latent time is represented by a color scale from 0 (the earliest latent time) to 1 (the latest latent time). (**C**) Left: UMAP display of stromal cell clusters that form a continuum of different cellular states (n=6376 cells). Colors correspond to different stromal clusters as in *Figure 2A*. Right: Heatmap constructed by the top 300 likelihood-ranked genes demonstrates gene expression dynamics along latent time. Colors on top of the heatmap correspond with cluster colors in UMAP (left). Key genes are highlighted by different colors on the right. Gene name colors correspond to different developmental stage transitions. Red: genes responsible for MSSC intra-cluster transition; orange: genes responsible for the transition to HAGEPs; light green: genes responsible for the transition to OCs; dark green: genes responsible for the transition to balanced progenitors and pre-osteoblasts.

The online version of this article includes the following figure supplement(s) for figure 3:

**Figure supplement 1.** Latent time, pseudotime, velocity confidence and expression dynamics of selected genes inferred by RNA velocity analysis.

(*Figure 3—figure supplement 1A–B*). Furthermore, generally high velocity confidences indicated that RNA velocity calculations were reliable (*Figure 3—figure supplement 1C*).

In addition to the identification of stromal differentiation directions and hierarchies, scVelo allowed us to detect potential key driver genes that govern cellular state transitions. Using likelihood-based computation, we instructed the program to identify the top 300 genes whose transcriptional dynamics correlated with cellular state transitions along the inferred latent time in the stromal continuum (*Figure 3C*, *Supplementary file 5*). A group of genes, including CSTA, B2M, CXCL12, and COL1A2 was found to drive the gradual and subtle cellular state transitions within the MSSC cluster (*Figure 3C*). Known adipogenic genes such as CEBPD (*Cao et al., 1991*) together with LEPR, CTSC, LMNA, KLF6, and EGR1 were identified to play important roles in the transition from MSSCs to HAGEPs (*Figure 3C*). As exemplified by CEBPD, EGR1, and LMNA (*Figure 3—figure supplement 1D–E*), gene transcriptional activity and expression level gradually increased from the MSSCs to HAGEPs. Furthermore, both novel drivers (NFKBIA, RBM3, TUBA1A, and DDX24), as well as reported genes such as ANXA1 (*Headland et al., 2015*), were identified to serve as critical genes in the transition from HAGEPs to OCs (*Figure 3C* and *Figure 3—figure supplement 1F*). VCAN expression has been demonstrated to correlate with bone formation in rats (*Nakamura et al., 2005*). Accordingly, VCAN was identified as one of the crucial genes for the transition from MSSCs to balanced progenitors and even to the pre-osteoblasts. Other potential novel driver genes along this path included CHL1 and TXNIP (*Figure 3—figure supplement 1G*) and NEAT1 and EIF5 (not shown). These genes demonstrated elevated expression levels in balanced progenitors and pre-osteoblasts.

In summary, RNA velocity analysis of the transcriptional dynamics allowed us to postulate stromal cell differentiation pathways and identify regulatory factors, including putative driver genes, that are potential key candidates to govern stromal cell fate commitment.

## Surface molecule profiling identifies marker combinations for the prospective isolation of functionally distinct stromal progenitor populations

Based on the expression patterns of stromal markers, osteochondrogenic genes, and fibroblastic markers, the nine stromal subsets could be further assigned to three main groups (*Figure 2* and *Figure 4—figure supplement 1A*). Group A cells (clusters 3, 5, 16, 38), which included MSSCs, HAGEPs, balanced progenitors, and pre-osteoblasts, showed high expression of stromal markers, adipogenic and osteogenic differentiation genes. Group B cells (clusters 29 and 23), annotated as OCs (*Figure 2*), demonstrated exclusive and specific expressions of osteochondro-lineage markers. Group C cells (clusters 6, 37, and 8), assigned as pre-fibroblasts (*Figure 2*), exhibited characteristic expression of a group of S100 genes, such as S100A4, S100A6, S100A8, and S100A9. These clear differences in gene expression between the groups pointed to potential functional differences. We therefore went on to identify suitable cell surface markers and marker combinations, respectively, that would allow for the prospective isolation and functional investigation of the three groups of cells.

As shown in *Figure 4A*, surface marker gene expression differed between groups A, B, and C.

Group B cells showed exclusive expression of NCAM1 and CD9, which were considerably lower expressed by group A and C cells (*Figure 4A*). Group C cells expressed several surface molecules which are conventionally considered as hematopoietic markers (CD52, CD37, PTPRC, and CD48). Thus, the lack of expression of NCAM1, CD9 and group C-specific antigens could be used to identify

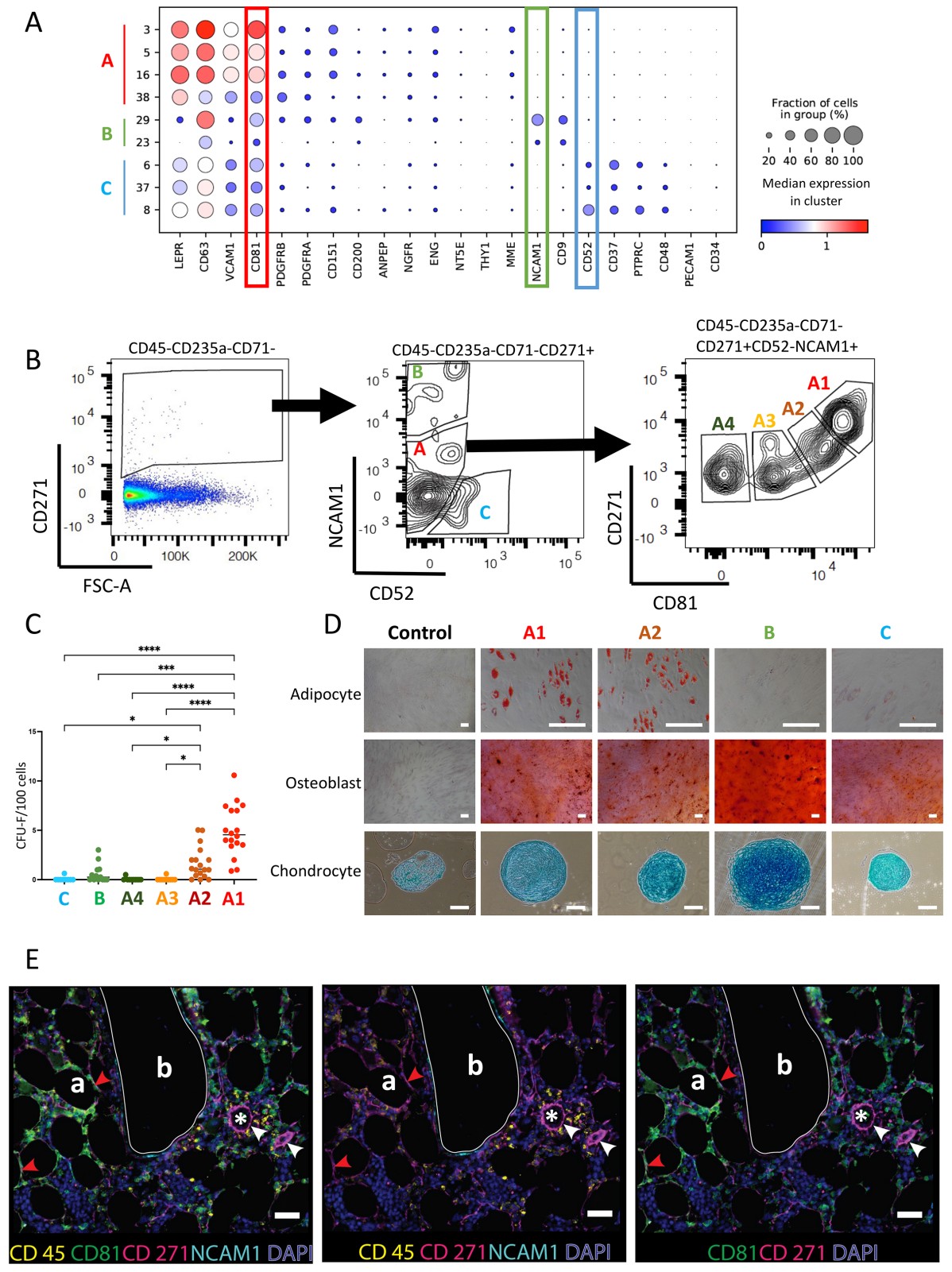

**Figure 4.** Stromal cell isolation based on data-driven gating strategies distinguishes cell subsets with different colony-forming capacities and differentiation capacities. (**A**) Dot plot of surface marker gene expression in different stromal cell clusters. Cluster numbers and corresponding stromal cell groups are indicated on the y-axis legend. Dot sizes represent the percentage of cells expressing a certain gene in each cluster and dot colors represent the scaled average expression of that gene. (**B**) FACS plots illustrating the gating strategy for the isolation of different stromal subsets.

*Figure 4 continued on next page*

*Figure 4 continued*

The displayed cell populations are indicated on top of the plot. Following exclusion of doublets, dead cells, CD45- and CD235a-expressing cells, CD45$^{low/-}$CD235a$^-$CD71$^-$CD271$^+$ cells (left panel) were gated based on CD52 and NCAM1 expression (middle panel). The resulting three populations were labelled as A-C (middle panel), corresponding to the stromal cell groups in (**A**) (A, CD52$^-$NCAM1$^-$; B, CD52$^-$NCAM1$^+$; C, CD52$^+$NCAM1$^-$). CD45$^-$CD235a$^-$CD71$^-$CD271$^+$CD52$^-$NCAM1$^-$ (group A) cells were further divided based on CD81 expression and four populations were identified (**A1–A4**) (right panel). A1, CD81$^{++}$; A2, CD81$^+$; A3; CD81$^{+/-}$; A4, CD81$^-$. (**C**) CFU-F frequencies of sorted stromal cell populations as shown in (**B**). Data are presented as individual data (dots) and median (horizontal lines) from independent experiments (n=3–6). Symbol colors and x-axis labels correspond to the cell population colors in (**B**). *: p<0.05; ****: p<0.0001 (Kruskal-Wallis test). (**D**) In vitro differentiation capacity of sorted stromal cell populations (as indicated in B) towards the adipogenic, osteoblastic, and chondrogenic lineage. Non-induction controls are shown in the left panel. Scale bars represent 200 μm. A representative set of pictures from a total of three independent experiments is shown. (**F**) Formalin-fixed, paraffin-embedded (FFPE) human BM slides were sequentially stained for DAPI (blue), CD45 (yellow), CD81 (green), CD271 (pink), and NCAM1 (cyan) and scanned with the OlympusVS120 slide scanner. Different staining combinations are shown as indicated under each picture to provide better visualization of individual staining obtained from the same FFPE slide. Scale bars represent 50 μm. Red arrows: CD271$^+$CD81$^{++}$ cells; white arrows: arteriolar walls; white lines: bone lining regions. Bone (**b**), adipocytes (**a**), and capillaries (*) are indicated.

The online version of this article includes the following figure supplement(s) for figure 4:

**Figure supplement 1.** Characterization of stromal cells based on gene expression, colony formation and in situ staining.

**Figure supplement 4.** Visualization of NCAM1, CD271, and SPP1 expression in bone marrow adipocyte regions by confocal microscopy.

group A cells, which could furthermore be separated into different subpopulations based on the differential expression of CD81.

Based on this, we designed FACS strategies for the isolation of group A (CD45$^{low/-}$CD235a$^-$CD71$^-$CD271$^+$NCAM1$^-$CD52$^-$), group B (CD45$^{low/-}$CD235a$^-$CD71$^-$CD271$^+$NCAM1$^+$CD52$^-$) and group C (CD45$^{low/-}$CD235a$^-$CD71$^-$CD271$^+$NCAM1$^-$CD52$^+$) cells (*Figure 4B*). As CD71 (TFRC) expression was barely detected in stromal clusters, CD71 was included as an additional exclusion marker to remove residual CD71-expressing erythroid cells which were present in the CD45$^{low/-}$CD235a$^-$ population (*Figure 1—figure supplement 1G*). Group A cells (CD45$^{low/-}$CD235a$^-$CD71$^-$CD271$^+$NCAM1$^-$CD52$^-$) were further separated into four subpopulations (A1-A4) based on CD81 expression, which corresponded to MSSCs (A1: CD81$^{++}$), HAGEPs (A2: CD81$^+$), balanced progenitors (A3: CD81$^{+/-}$), and pre-osteoblasts (A4: CD81$^-$) (*Figure 4B*). Quantitative real-time PCR of sorted cells demonstrated that the expression levels of CD81 in A1 and A2 were higher than A3 and A4, indicating these four phenotypically defined subsets reflected the transcriptionally identified clusters in group A (*Figure 4—figure supplement 1B*).

Functional analysis of sorted cells showed that the MSSC population (A1: CD45$^{low/-}$CD235a$^-$CD71$^-$CD271$^+$NCAM1$^-$CD52$^-$CD81$^{++}$) had the highest potential to form fibroblastic colonies (colony-forming units, fibroblast: CFU-F) and multi-lineage differentiation capacity toward osteoblasts, adipocytes, and chondrocytes (*Figure 4C–D*, *Figure 4—figure supplement 1C*). Although the CFU-F frequency of HAGEPs (A2: CD45$^{low/-}$CD235a$^-$CD71$^-$CD271$^+$NCAM1$^-$CD52$^-$CD81$^+$) was lower compared to MSSCs, they still gave rise to a significant fraction of colonies and showed full in vitro differentiation capacities (*Figure 4C–D*, *Figure 4—figure supplement 1C*).

Colony-forming capacities of balanced progenitors (A3: CD45$^{low/-}$CD235a$^-$CD71$^-$CD271$^+$NCAM1$^-$CD52$^-$CD81$^{+-}$), pre-osteoblasts (A4: CD45$^{low/-}$CD235a$^-$CD71$^-$CD271$^+$NCAM1$^-$CD52$^-$CD81$^-$) and OCs (B: CD45$^{low/-}$CD235a$^-$CD71$^-$CD271$^+$NCAM1$^+$CD52$^-$) were considerably lower in comparison with MSSCs and HAGEPs (*Figure 4C* and *Figure 4—figure supplement 1C*). Pre-fibroblasts (group C) showed the lowest potential to form fibroblastic colonies (*Figure 4C* and *Figure 4—figure supplement 1C*). Furthermore, the adipogenic potential of pre-fibroblasts was markedly reduced and the pre-fibroblast-derived chondrocyte pellets were smaller compared to the other groups while osteogenic differentiation was normal (*Figure 4D*), thus indicating a limited differentiation potential of this population. OCs (group B) demonstrated an enhanced capacity to differentiate towards osteoblasts and chondrocytes, however, at the expense of adipogenic potential (*Figure 4D*). These findings suggest a skewed differentiation potential of OCs, which is consistent with the lower expression of adipogenic markers and elevated expression of osteochondrogenic markers (*Figure 2C and E*). The in vitro differentiation capacities of balanced progenitors (A3) and pre-osteoblasts (A4) could not be assayed due to insufficient proliferation upon culture expansion.

Taken together, the data-driven FACS gating strategy allowed us to isolate distinct stromal subsets for functional investigation. Both MSSCs and HAGEPs demonstrated considerably higher in vitro colony-forming capacities in comparison with balanced progenitors, OCs, pre-osteoblasts, and pre-fibroblasts. While MSSCs and HAGEPs exhibited full multi-differentiation potentials, pre-fibroblasts had only limited differentiation potentials. OCs showed enhanced osteoblastic and chondrogenic differentiation capacities and compromised adipogenic potential, which is consistent with their differentiation marker expression profiles (*Figure 2*).

## In-situ localization of stromal cell subpopulations

Based on the markers identified above, we used CD271 (NGFR), CD81, NCAM1, and CD45 to identify stromal cells (CD271$^+$) and to investigate the in-situ localization of MSSCs (CD271$^+$CD81$^{++}$) as well as OCs (CD271$^+$NCAM1$^+$) using bone marrow biopsies from hematologically normal donors. As shown in *Figure 4E* and *Figure 4—figure supplement 1D–E*, CD271-expressing cells were found in perivascular regions, endosteal regions, and throughout the stroma. We observed that perivascular cells surrounding the capillary endothelium and larger vessels primarily expressed CD271 alone. The bone-lining cells proximal to the surface of trabecular bone showed expression of both CD271 and NCAM1, confirming the osteochondrogenic nature of OCs, which correlated very well with the predicted endosteal localizations of murine osteoblasts and chondrocytes (*Baccin et al., 2020*). In contrast to the CD271/NCAM1 double-positive cells, CD271/CD81 double-positive cells were localized in bone marrow stromal regions, including both peri-adipocytic and perivascular regions with a considerable fraction of CD271/CD81 cells being located in close proximity or encircling adipocytes. Co-staining with CD45 confirmed that perivascular, periadipocytic, and bone-lining cells were CD45 negative. The results from the stromal cell in situ localization analysis by scanning microscopy were confirmed by detailed confocal microscopy analysis as illustrated in *Figure 4—figure supplements 2–16*. These results thus demonstrated the distinct anatomical localizations of MSSCs and OCs, which confirms and extends our previous findings on differently localized putative hematopoietic niche cells (*Tormin et al., 2011*) and point to possible divergent physiological functions of the different stromal subsets.

## In silico prediction of cell-cell interactions in the human bone marrow microenvironment

It has been proposed that different niches exist for different types of hematopoietic stem and progenitor cells. We therefore went on to study the interactions between the different human BM stromal cell populations identified herein and hematopoietic cells based on ligand-receptor (LR) expression using CellPhoneDB (*Efremova et al., 2020*) complemented by analysis of additional published LR pairs for erythroid cells (*Kleven et al., 2018*).

As shown in *Figure 5A* and *Figure 5—figure supplement 1A*, we identified a wide range of interactions between stromal cells and hematopoietic cells. Generally, cells belonging to the three stromal groups (A-C) were predicted to have multiple interactions with a large number of hematopoietic cell types, including HSPCs and more mature hematopoietic cells (*Figure 5A* and *Figure 5—figure*

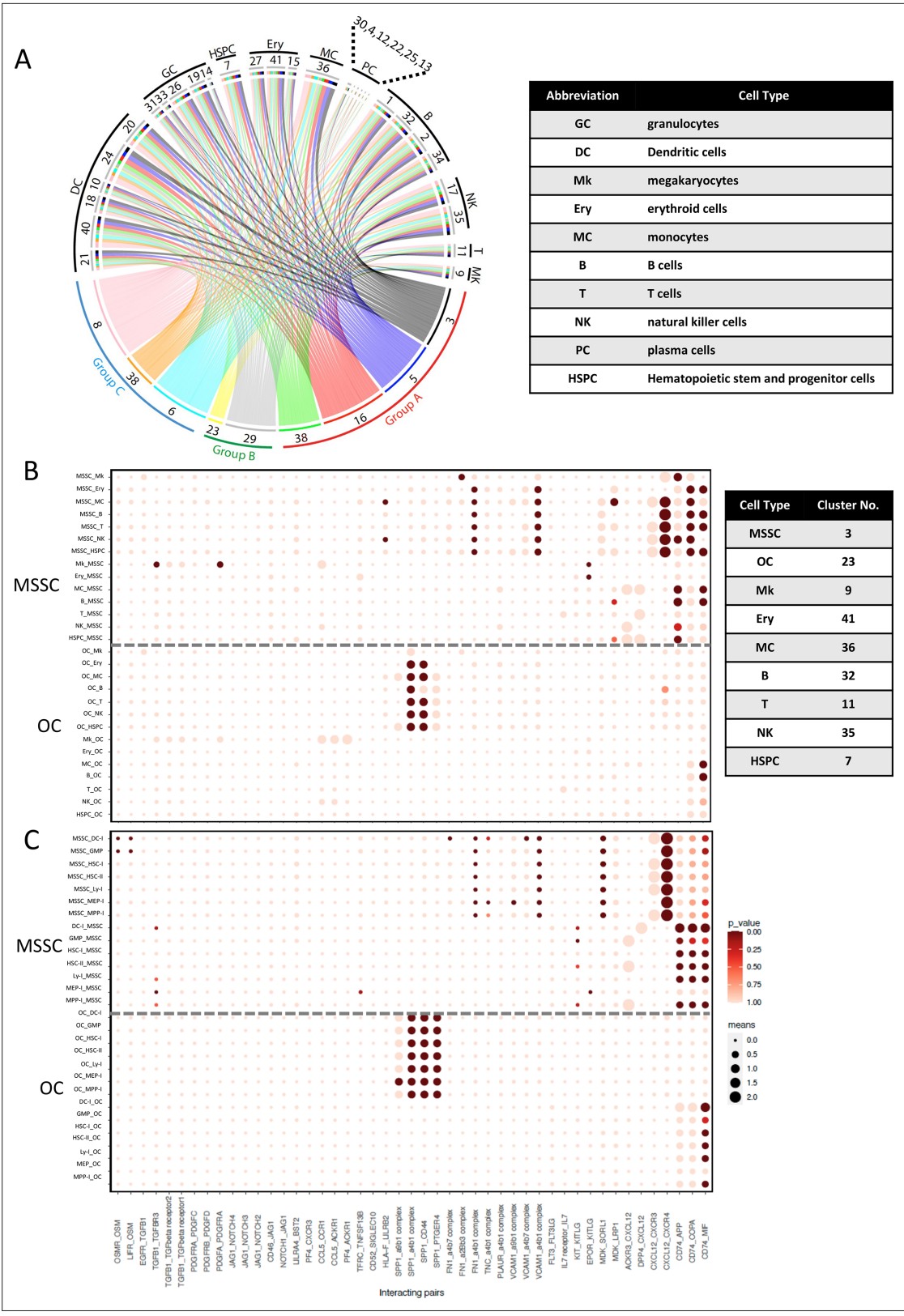

**Figure 5.** Cell-cell interaction between stromal cells and hematopoietic cells in human bone marrow. (**A**) Circos plot visualization of cell-cell interaction between bone marrow stromal cells and different hematopoietic cell types. Colored lines connect stromal clusters with hematopoietic clusters based on LR expression analysis. Line colors indicate cell type and line thicknesses correspond to the number of interacting LR pairs. Cluster numbers and corresponding cell types and stromal groups are indicated. Hematopoietic cell type abbreviations are listed in the table. (**B and C**) Overview

*Figure 5 continued*

of selected ligand-receptor pairs between MSSCs and OCs, respectively, and different hematopoietic cell types from this dataset (**B**) and more defined hematopoietic progenitors from another study (**C**). The y-axis label indicates the pair of interacting cell clusters ('cluster X_cluster Y' indicates cluster X interaction with cluster Y by ligand-receptor expression). The x-axis indicates the interacting receptor/ligand (R/L) or ligand/receptor pairs, respectively ('molecule L_molecule R', molecule L interacts with molecule R). The means of the average expression level of the interacting molecules are indicated by circle size (adjacent lower scale bar). P values are indicated by circle color and correspond to the upper scale bar. Cluster numbers of the hematopoietic cell types are listed in the table. The dashed line separates MSSC from OC interactions. DC-I, dendritic cell progenitor cluster I; GMP, granulocyte-macrophage progenitor cluster I; HSC-I, hematopoietic stem cell cluster I; HSC-II, hematopoietic stem cell cluster II; Ly-I, lymphoid progenitor cluster I; MEP-I, megakaryocyte–erythroid progenitor cluster I; MPP-I, multipotent progenitor cluster I.

The online version of this article includes the following figure supplement(s) for figure 5:

**Figure supplement 1.** Cell-cell interactions inferred by CellPhoneDB.

**Figure supplement 2.** UMAP illustration of selected ligands and receptors involved in cell-cell interaction.

**Figure supplement 3.** Cell-cell interaction between different clusters and functional assays.

---

*supplement 1A*). Interestingly, only a few LR pairs were identified between stromal cells and plasma cells (*Figure 5A*).

To study the functional relationships between stromal and hematopoietic cells in more detail and to identify subset-specific interactions, we plotted selected ligand-receptor pairs that are known to be relevant for hematopoietic-stromal crosstalk (*Figure 5B*). Selected LR pairs, including essential cytokines and established hematopoiesis-supporting molecules were analyzed for effects from stromal cells on hematopoiesis and vice versa, including stromal subset-specific interactions, intra-, and inter-stromal cell interactions, and other potentially important interactions (*Figure 5B*, *Figure 5—figure supplement 1B*, *Figure 5—figure supplement 3B*).

## Stromal cell regulation of hematopoiesis

Regarding the regulatory and supportive effects of stromal cells on hematopoiesis, we found that group A and C stromal subsets had the potential to interact with essentially all hematopoietic cell types through similar pathways. CXCL12 was highly expressed by group A and C stromal cells, especially the MSSC population (*Figure 6A*), and CXCL12-CXCR4 crosstalk was detected in almost all hematopoietic clusters except for erythroid progenitors (*Figure 5B*, *Figure 5—figure supplement 1B*, and *Figure 6A*). Furthermore, large dot sizes and high statistical power indicated that CXCL12-mediated interactions represent one of the major interaction mechanisms between these stromal cells and hematopoietic cells (*Figure 5B*, *Figure 5—figure supplement 1B*). In contrast, CXCL12 interactions with CXCR3 and DPP4 were less significant (*Figures 5B and 6A*).

KITLG (SCF)-involving stroma-hematopoiesis interactions were also identified. However, these interactions were not as strong as those involving CXCL12 (*Figure 5—figure supplement 2A*). Interestingly, MDK, a neurite growth factor expressed by stromal cells, showed interactions with the HSPCs, lymphoid lineage progenitors (B, T, and NK), and monocytes through LRP1 or SORL1 (*Figure 5—figure supplement 2B*), which has thus far only been described in mouse fetal liver (*Gao et al., 2022*). Furthermore, other cytokines known to be important for the maintenance of CD34[+] HSPC cells include FLT3-ligand (FLT3LG) and thrombopoietin (THPO). However, whereas expression of FLT3 was detected in several hematopoietic cell types including HSPCs, dendritic cells and monocytes (*Figure 5—figure supplement 2C*), THPO and MPL expression were barely detected (*Figure 5—figure supplement 2D*). JAG1 expression was highly enriched in stromal cells while NOTCH1, 2 and 4 expressions were detected in hematopoietic cells, suggesting that activation of Notch signaling is involved in stromal-hematopoietic interplay (*Figure 5—figure supplement 2E*). All three stromal groups were furthermore predicted to interact with T cells through the IL7-IL7R RL combination, indicating a T cell supporting mechanism by stromal cells (*Figures 5B and 6B*, *Figure 5—figure supplement 1B*).

In addition to soluble factor-mediated interactions, in silico analysis also identified direct cell-to-cell communication between group A and C stromal cells and hematopoietic cells mediated by VCAM1 (*Figure 6C*), FN1, PLAUR and TNC, some of which have also been identified in a previously reported study on murine cells (*Mende et al., 2019*). These results indicated that group A and C cells have

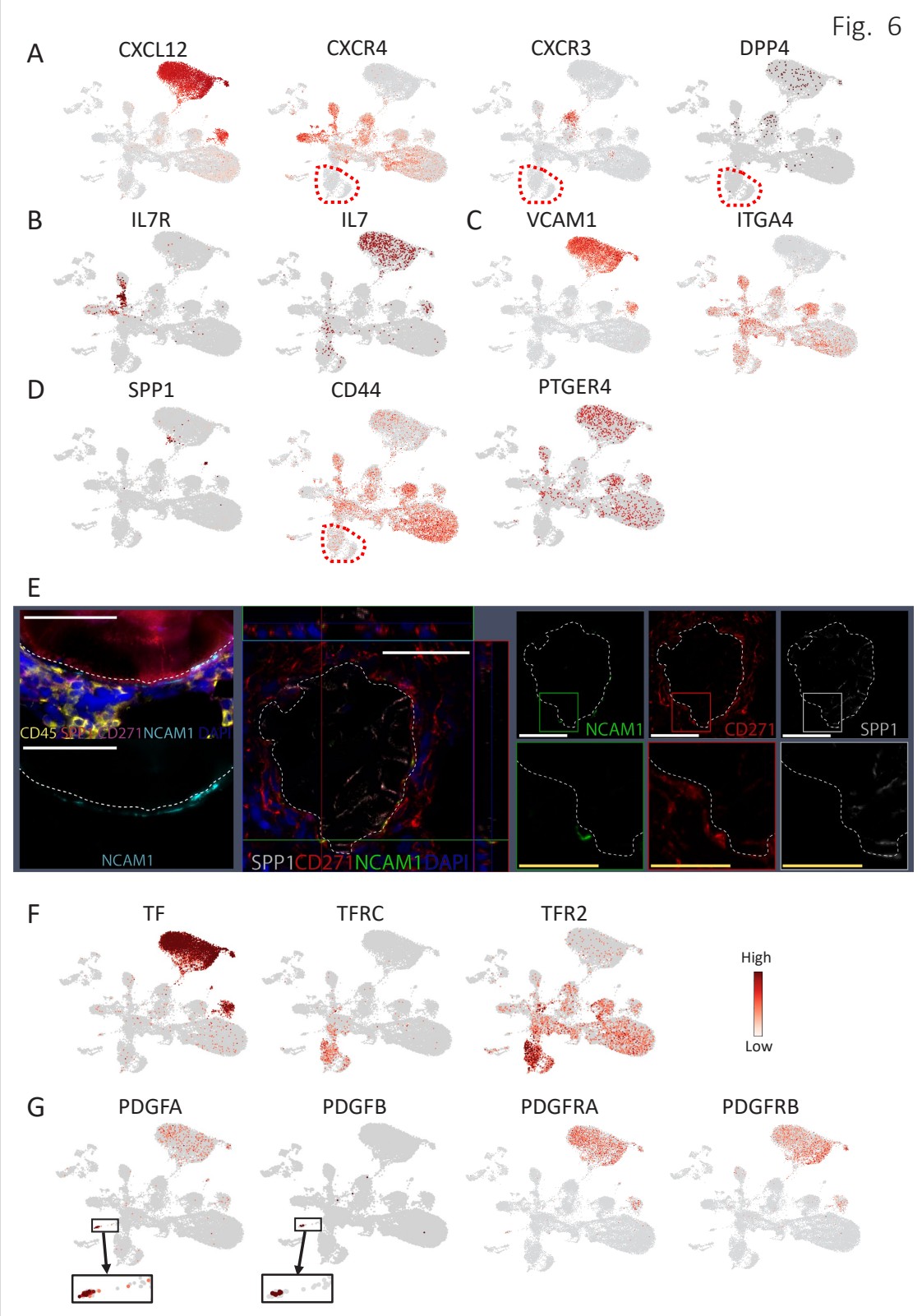

**Figure 6.** Expression of selected ligand and receptor pairs involved in cell-cell interaction between stromal cells and hematopoietic cells in human bone marrow. (**A–D, F–G**) UMAP (as in *Figure 1B*) illustration of the normalized expression of selected ligand and receptor gene pairs. Red dashed lines in figures A, B and D mark the erythroid clusters. The blow-up shown in (**G**) is to better visualize PDGFA- and PDGFB-expressing cells. (**E**) Left panel: Formalin-fixed, paraffin-embedded (FFPE) human BM slides were sequentially stained for DAPI (blue), CD45 (yellow), SPP1 (red), CD271

*Figure 6 continued on next page*

Figure 6 continued

(pink), and NCAM1 (cyan) and scanned with an OlympusVS120 slide scanner. Left upper image: all markers are shown; left lower image: just NCAM1 channel is shown. Middle panel: confocal laser scanning analysis of BM biopsies co-stained with CD271 (red), NCAM1 (green), SPP1 (white), and DAPI (blue) presented as 3D orthographic cross-section view. Right panel: Single staining channel data for CD271 (red), NCAM1 (green), SPP1 (white), and corresponding blow-ups for indicated areas. NCAM1, SPP1 and CD271(NGFR). White scale bars represent 50 µm and yellow scale bars represent 25 µm. White dashed lines indicate the trabecular bone surface lining regions.

the potential to support hematopoietic cells via direct cell-to-cell contact as well as hematopoiesis-supporting cytokines.

In contrast to the strong contribution of CXCL12-mediated interactions between group A and C stromal cells and hematopoietic cells, SPP1-mediated interactions were predominant in the crosstalk between group B stromal cells (OCs) and hematopoietic cells (except for megakaryocytes), indicating a stromal group-specific interaction (*Figure 5B* and *Figure 5—figure supplement 1B*). Interacting partners of SPP1-expressing OCs were erythroid cells, lymphoid progenitors (B, T, NK progenitors) and CD34-enriched HSPC cluster which expressed SPP1 binding partners such as CD44 and PTGER4 (*Figures 5B and 6D* and *Figure 5—figure supplement 1B*). SPP1-involving integrin alpha 4 beta 1 (a4b1) and alpha 9 beta 1 (a9b1) complexes could also play important roles in the interaction of OCs with a number of hematopoietic cells. Consistent with this, FACS analysis demonstrated that SPP1 expression was higher in OCs in comparison with MSSCs (*Figure 5—figure supplement 2F*). More-over, in situ staining demonstrated that SPP1-expressing cells were exclusively colocalized endos-teally with NCAM1$^+$ OCs (*Figure 6E*), pointing to a localization-specific regulation. Importantly, the CXCL12- and SPP1-mediated stromal cell type-specific interactions detected in our dataset were furthermore confirmed when performing R/L analysis of our stroma cell data with well-defined human BM hematopoietic stem and progenitor clusters obtained from a recently published study (*Sommarin et al., 2021*; *Figure 5C*).

Taken together, these data demonstrated that the most predominant interactions were CXCL12- and SPP1- mediated interactions for group A/C stromal cells and OCs, respectively, indicating that regulation of hematopoiesis is stromal group and location-specific.

## Erythroid cell-specific interactions

Unlike the majority of the hematopoietic cells in our dataset, the interplay between erythroid cells and group A and C stromal cells was not dependent on CXCL12-involving interactions as indicated by lacking expression of CXCL12 receptors in erythroid clusters (*Figure 6A*). CellPhoneDB analysis demonstrated that while group A and C stromal cells had the potential to interact with the erythroid cells through VCAM1-, FN1-, TNC- and PLAUR- involving complexes, the interactions between OCs (group B) and erythroid cells were mainly mediated by SPP1-CD44 (*Figures 5B and 6D* and *Figure 5—figure supplement 1B*). In addition to that, we found that TF (transferrin) was exclusively expressed by all stromal clusters and its corresponding receptors, TFRC and TFR2 were highly expressed by erythroid progenitors (*Figure 6F*).

## Hematopoietic regulation of stromal cells

Aside from the supportive and regulatory role of stromal cells for hematopoiesis, it is well described that regulatory signals originating from hematopoietic cell types can affect stromal cells (*Baksh et al., 2005*). Accordingly, we found that the crucial stromal cell growth factors PDGFs (PDGFA, PDGFB, and PDGFC) were highly expressed by megakaryocytes while their corresponding receptors (PDGFRA and PDGFRB) were predominantly expressed by stromal cells (*Figure 6G*), suggesting interactions through these axes (*Figure 5B*, *Figure 5—figure supplement 1B*). Our analysis also predicted that hema-topoietic cells control stromal cells via the TGFB1-EGFR and TGFB1-TGFBR1/2/3 axes (*Figure 5B*, *Figure 5—figure supplement 1B*, *Figure 5—figure supplement 2G*). Furthermore, Oncostatin M (OSM) expression was found in various hematopoietic cell types, indicating a possible regulation of stromal cells through its receptors, OSMR and LIFR (*Figure 5B*, *Figure 5—figure supplement 1B*, *Figure 5—figure supplement 2H*).

Together, these data indicate that the regulatory effects between stromal cells and hematopoietic cells are bidirectional, which is consistent with the increased fibroblastic colony size in the presence of CD45$^+$ hematopoietic cells in CFU-F assays (*Figure 5—figure supplement 2I*).

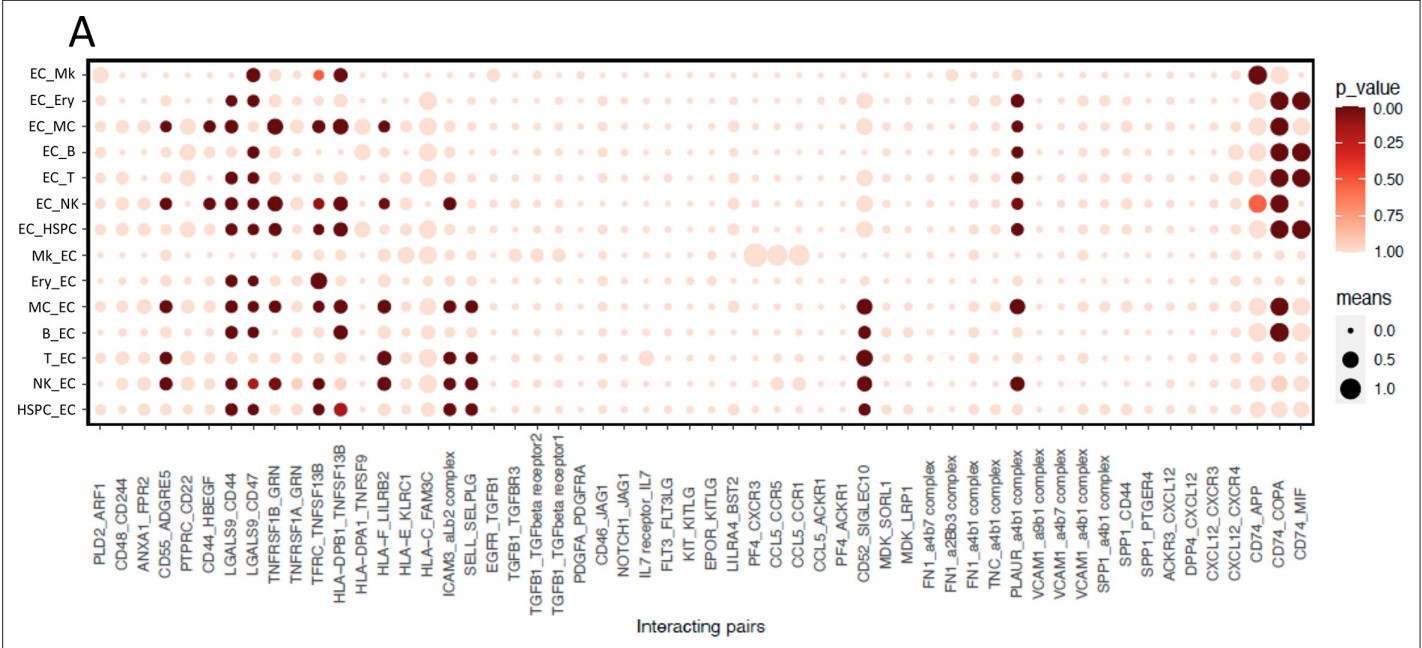

**Figure 7.** Cell-cell interaction between the endothelial cluster and hematopoietic cells in human bone marrow. (**A**) Overview of selected ligand-receptor pairs between the endothelial cluster (cluster 28) and different hematopoietic cell types. The y-axis label indicates the pair of interacting cell clusters ('cluster X_cluster Y' indicates cluster X interaction with cluster Y by ligand-receptor expression). The x-axis indicates the interacting receptor/ligand (R/L) or ligand/receptor pairs, respectively ('molecule L_molecule R', molecule L interacts with molecule R). The means of the average expression level of the interacting molecules are indicated by circle size (adjacent lower scale bar). P values are indicated by circle color and correspond to the upper scale bar. Cluster numbers of the hematopoietic cell types are listed in the table in **Figure 5B**.

### Inter- and intra-stromal signaling

As shown in **Figure 5—figure supplement 3A**, CFU-F frequencies were significantly reduced when stromal cells were cultured under single-cell conditions in comparison with bulk-cultured cells, suggesting the presence of inter- and intra-stromal signaling among stromal cells. Indeed, multiple stimulatory pathways were operative between different stromal subsets as well as within the same stromal cell population (**Figure 5—figure supplement 3B**). Among them, FGF receptor-mediated signaling and various collagen-involving pathways were identified that reflected interactions between different stromal cells (**Figure 5—figure supplement 3C**). In addition, CXCL12-, MDK-, GRN-, NRP1-, and BST2- mediated interactions were also found to significantly contribute to the regulatory mechanisms of stromal cells (**Figure 5—figure supplement 3B–D**).

In summary, our data showed all three stromal groups demonstrate the potential to interact with a wide range of hematopoietic cells. According to this analysis, group A and C cells were predicted to communicate with hematopoietic cells mainly through CXCL12-, VCAM1-, FN-, and MDK-involving pathways, whereas group B cells have the potential to interact with hematopoiesis mainly via SPP1-mediated pathways. Furthermore, regulation was predicted to be bidirectional, that is stromal cells are regulated by hematopoietic cells through PDGF-, OSM-, and TGFB1-mediated signaling. Additionally, multiple intra- and inter-stromal pathways were identified. Our in silico analysis of scRNAseq data of both stromal and hematopoietic clusters thus predicted the presence of a complicated interaction network in human bone marrow microenvironment.

### Cell-cell interaction between endothelial cluster and hematopoietic clusters

As endothelial cells are an important niche component, we also compared the interaction pattern between endothelial cells and hematopoietic cells with that of MSSCs. As illustrated in **Figure 7A**, the endothelial cluster had a broad interaction spectrum with all hematopoietic clusters. In addition to the endothelial-specific SELL-SELPLG and ICAM3-aLb2 complex pathways, it was predicted that the endothelial cluster interacted with hematopoietic clusters mainly through HLA molecules-, TNFRSF family members- and LGALS9-mediated interactions. However, endothelial cluster interaction seemed

to be less specific compared with stromal cells. Unlike stromal cells, CXCL12-mediated interactions were not identified as the main signaling pathways in the crosstalk between endothelial cells and hematopoietic cells, indicating that endothelial cells and stromal cells communicate with hematopoietic cells through different mechanisms.

## Discussion

The BM microenvironment plays a critical role in regulating hematopoiesis. Recent landmark studies have illustrated the molecular complexity of the murine BM microenvironment (*Baryawno et al., 2019*; *Tikhonova et al., 2019*; *Baccin et al., 2020*). However, the exact definition of the cell populations that form the BM stroma in humans remains elusive. In this study, based on single-cell RNA sequencing technology, we gained important insights into the stromal components of this important organ including their developmental and functional roles in hematopoiesis.

Several recent studies using scRNAseq approaches provided the first important insights into the complexity of the human BM and the potential functional roles of BM stromal cells (*de Jong et al., 2021*; *Triana et al., 2021*; *Wang et al., 2021*). A first overview of the cellular composition of the nonhematopoietic cells in human BM was provided by a recently published single-cell proteo-genomic reference map (*Triana et al., 2021*) and a single-cell transcriptomic map of non-hematopoietic cells (*de Jong et al., 2021*). However, due to the low frequencies of the stromal stem/progenitor cells, it has been difficult to investigate the potential heterogeneity of the human BM stromal cell compartment even when employing single cell approaches. Enrichment strategies represent a possible approach for a detailed analysis of the transcriptional diversity of BM stromal cells, as recently reported by *Wang et al., 2021*. However, in their study, CD271+ cells from femoral shafts from old patients with osteoarthritis and osteoporosis were used, which might not reflect transcriptomic profiles of normal stroma cells from hematologically active bone marrow (*Wang et al., 2021*). Because of the extremely low frequency of stromal cells in human BM, we chose a sorting strategy that also included CD45low cells to ensure that no stromal cells were excluded from the analysis. Furthermore, we also added sorted CD271+ cells, which are highly and exclusively enriched for stromal stem/progenitor cells (*Tormin et al., 2011*). Therefore, despite the expected presence of hematopoietic cells in our dataset, this approach ensured that stromal cell resolution was substantially increased.

Following exclusion of the hematopoietic clusters, we identified primarily nine stromal clusters which could further be assigned to six stromal cell types, that is MSSCs, HAGEPs, balanced progenitors, OCs, pre-osteoblasts, and pre-fibroblasts. Of note, corresponding murine stromal cell types have been reported for most of these cell types (MSSCs, HAGEPs, OCs, pre-osteoblasts, and pre-fibroblasts), whereas the cluster annotated as balanced progenitors has not been described either in mice or in humans (*Baryawno et al., 2019*; *Tikhonova et al., 2019*).

Additionally, we identified three non-hematopoietic cell clusters which contained KRT5- (cluster 0) and neuronal gene-expressing cells (cluster 39), respectively, as well as endothelial cells. KRT5 is a keratin family member and is widely used as a basal cell marker. However, KRT5 expression has also been reported for several immune cell types, such as plasmacytoid dendritic cells and NK cells. As cells in our analysis did not express typical immune cell genes, further characterization of this cluster needs to be performed in future studies. Multiple genes indicating the presence of different cell types were identified in cluster 39. The genes identified with the highest fold changes included several neuronal markers (NEUROD1, CHGB, ELAVL2, ELAVL3, ELAVL4, STMN2, INSM1, ZIC2, NNAT) (*Supplementary file 4*). Due to the heterogeneity of this cell composition, this cluster was therefore annotated as neuronal cell-containing cluster.

Thus, this analysis allowed us to dissect the human bone marrow non-hematopoietic cell compartment at a thus far unreached molecular level, to resolve the stromal cell heterogeneity and generate a detailed transcriptional fingerprint of distinct stromal populations.

Furthermore, we were able to define phenotypically and functionally distinct stromal subsets, which were predicted to be organized in a hierarchical manner according to RNA velocity analysis. We identified MSSCs as the most primitive population, which is in accordance with published murine and human stromal stem cell studies (*Zhou et al., 2014*; *Baryawno et al., 2019*; *Wolock et al., 2019*). Pre-fibroblasts, pre-osteoblasts, and OCs were placed downstream of MSSCs in the developmental hierarchy, thus confirming the reliability of the velocity analyses. By using a likelihood-based computation, we also detected potential key driver genes that are candidates to govern the transition between

different cellular states. Identified genes included both novel and established transcription factors and certainly, these data provide the basis for future important mechanistic studies of the dynamic processes associated with cellular state transition and fate choices.

Self-renewing human skeletal stem cells (hSSC) with osteogenic and chondrogenic potential have been previously identified in human fetal and adult bones based on the phenotype PDPN⁺CD146⁻CD73⁺CD164⁺ (*Chan et al., 2018*). However, the identity of a tri-lineage multipotent stromal progenitor in adult human bone marrow was not reported. Herein, we were able to provide a phenotypical definition that allowed for the faithful identification and prospective isolation of molecularly defined stromal stem and progenitors from hematopoietically active adult bone marrow. These cells have thus far remained incompletely characterized and our data thus complement and extend reported phenotype definitions of MSCs from bone marrow and skeletal tissues (*Jones et al., 2002*; *Quirici et al., 2002*; *Gronthos et al., 2007*; *Sacchetti et al., 2007*; *Delorme et al., 2008*; *Li et al., 2016*; *Chan et al., 2018*). The phenotypic MSSCs defined in this study also showed expression of previously reported stromal surface markers such as CD51 (*Pinho and Frenette, 2019*) and CD146 (*Tormin et al., 2011*; data not shown). Besides these surface markers, scRNAseq also identified VCAN as a potentially important stromal gene based on its high expression in stromal clusters. However, as an extracellular matrix protein, FACS analysis of cellular VCAN expression can only be achieved based on its intracellular expression after fixation and permeabilization. Additionally, VCAN is also expressed by monocytes (cluster 36). Therefore, VCAN is not an optimal marker to isolate viable stromal cells.

Furthermore, we performed a preliminary analysis which indicated possible age- and gender-related stromal differences as also reported for murine cells (*Singh et al., 2016*; *Maryanovich et al., 2018*; *Aguilar-Navarro et al., 2020*). However, this potentially interesting finding needs to be confirmed by additional experimentation in forthcoming studies.

In mice, elegant studies using genetic approaches involving in-vivo labelling of cell types and ablation of candidate niche cells and niche factors have provided detailed insight into the functional roles of stromal cells in regulating hematopoiesis (*Pinho and Frenette, 2019*). Comparable studies in human are impossible to perform and analysis of human stromal cells function has been limited to in-vitro models. We therefore investigated cellular interactions based on ligand-receptor interactions and found a broad and complicated crosstalk network, which is likely to reflect – at least in part – the in-situ situation as data were generated with directly isolated cells. We found that stromal cells generally showed hematopoiesis supporting potential, which is consistent with previously published in-vitro coculture data (*Li et al., 2014*; *Li et al., 2020*) and which we confirmed in co-cultures with BM CD34⁺ cells and MSSC- or OC-derived stromal feeder cells (*Figure 5—figure supplement 3E*). Furthermore, whereas all stromal cell groups showed the potential to interact with almost all hematopoietic cells, cell-cell interaction mechanisms differed considerably between different groups of stromal cell groups, that is between group A/C cells which included the MSSCs and group B cells (OCs). Whereas the former demonstrated the potential to interact with hematopoiesis through cytokines such as CXCL12, IL7, KITLG, MDK, and TF as well as adhesion molecules providing direct cell-to-cell contact (e.g. VCAM1), SPP1-mediated interactions were predicted to be the main pathways for the latter. Although we have not formally examined the expression of the receptors for CXCL12 and SPP1 on hematopoietic cells in the current study, expressions of CXCR4 (CXCL12 receptor) and CD44 (SPP1 receptor) on human BM hematopoietic cells has been demonstrated previously (*Ishii et al., 1999*; *AbuSamra et al., 2017*).

These results thus indicated that hematopoietic cells were maintained by different stromal populations through diverse but nevertheless stromal cell-specific pathways. Taking furthermore into account that SPP1-expressing OCs were located endosteally and that CD271 single positive stromal cells including MSSCs were localized in the perivascular and stromal regions, respectively, suggested the possibility that different stromal cells provide specialized niches for hematopoietic cells in different locations. Our data thus confirm and extend previous reports describing that endosteal and perivascular niches exist in human BM (*Tormin et al., 2011*; *Pinho et al., 2013*), and conceptually support a model where different hematopoietic cell types are differentially regulated by distinct niche milieus which are composed of specific cytokine-producing stromal cells. Finally, in silico ligand-receptor interaction predictions suggested that endothelial cells and stromal cells showed the potential to communicate with hematopoiesis through different mechanisms, which is certainly an interesting finding that needs to be further addressed in future studies.

In this study, we used human BM aspiration samples in which bone-lining or bone-attaching cells are most likely underrepresented. Our experiments could therefore ideally be complemented by studies on fresh bone marrow biopsy samples from healthy individuals, which however were not available to us, and which might be difficult to acquire because of ethical considerations. In our study, tri-lineage differentiation capacities of stromal clusters were evaluated using standard in-vitro assays, which may not necessarily reflect the bona fide stromal cell potentials. Thus, these data need to be complemented in follow-up studies by in vivo differentiation studies, for example using humanized BM ossicles. The potential key genes that govern the stromal cell fate commitment were predicted using likelihood-based computation and the stroma-hematopoiesis crosstalk mechanisms were inferred based on ligand-receptor interactions. The identified key genes and interactions will need to be validated in future experiments, for example using in vitro engineered human BM models or in vivo humanized BM ossicles (*Dupard et al., 2020*). While the CellPhoneDB analysis predicted the communication networks between stromal and hematopoietic cells, this approach infers potential interactions using transcriptomics data without considering the spatial proximity of the cells and functional relevance of the interaction. A more comprehensive overview of cellular communication is anticipated when combined with spatial localization analysis and functional assays in future studies. Despite these limitations, we believe that our results provide the basis for a better understanding of the cellular complexity of the human BM microenvironment and put a new light on the current concept that specialized niches exist for distinct types of hematopoietic stem and progenitor cells (*Morrison and Scadden, 2014*).

## Acknowledgements

The authors thank Helene Larsson and Anna Jonasson for their help to collect bone marrow samples and the Lund Stem Cell Center FACS facility personnel for technical assistance. We would also like to thank Dr. Paul Bourgine for critically reading the manuscript. Funding: This work was supported by funds from the StemTherapy Program, the Swedish Cancer Foundation, the Swedish Childhood Cancer Foundation, the Swedish Bloodcancer Association (Blodcancerförbundet), Foundation Siv-Inger and Per-Erik Anderssons minnesfond, John Persson Foundation, ALF (Government Public Health Grant), and the Skåne County Council Research Foundation.

## Additional information

### Funding

| Funder | Grant reference number | Author |
|---|---|---|
| Swedish Cancer Foundation | 20-1163PjF 01H | Stefan Scheding |
| Swedish Childhood Cancer Foundation | PR2018-0078 | Stefan Scheding |
| Swedish Childhood Cancer Foundation | PR2021-0065 | Stefan Scheding |
| Swedish Bloodcancer Association | | Stefan Scheding |

The funders had no role in study design, data collection and interpretation, or the decision to submit the work for publication.

### Author contributions

Hongzhe Li, Conceptualization, Data curation, Formal analysis, Supervision, Validation, Investigation, Visualization, Methodology, Writing – original draft, Project administration, Writing – review and editing; Sandro Bräunig, Data curation, Formal analysis, Validation, Investigation, Visualization, Methodology, Writing – original draft, Writing – review and editing; Parashar Dhapolar, Data curation, Validation, Investigation; Göran Karlsson, Supervision; Stefan Lang, Data curation, Formal analysis, Methodology; Stefan Scheding, Conceptualization, Resources, Supervision, Funding acquisition, Writing – original draft, Project administration, Writing – review and editing

## Author ORCIDs
Hongzhe Li http://orcid.org/0000-0001-7788-878X
Sandro Bräunig http://orcid.org/0000-0002-4418-0854
Stefan Scheding http://orcid.org/0000-0002-8005-9568

## Ethics
Human bone marrow (BM) cells were collected at the Hematology Department, Skåne University Hospital Lund, Sweden, from consenting healthy donors. The use of human samples was approved by the Regional Ethics Review Board in Lund, Sweden.

## Decision letter and Author response
Decision letter https://doi.org/10.7554/eLife.81656.sa1
Author response https://doi.org/10.7554/eLife.81656.sa2

## Additional files

### Supplementary files
- MDAR checklist
- Supplementary file 1. Donor information.
- Supplementary file 2. Cluster annotation.
- Supplementary file 3. Cell numbers and percentages of each cluster in each sample.
- Supplementary file 4. DE (differentially expressed) genes in each cluster.
- Supplementary file 5. List of the 300 top-ranked likelihood genes shown in *Figure 3C*.
- Supplementary file 6. List of ligand-receptor interactions in scRNAseq dataset as identified by CellPhoneDB analysis.
- Supplementary file 7. Stromal cluster summary.

### Data availability
The scRNA-seq matrix data generated in this study have been deposited in the GEO database (GSE190965).

The following dataset was generated:

| Author(s) | Year | Dataset title | Dataset URL | Database and Identifier |
|---|---|---|---|---|
| Li H, Lang S | 2023 | Transcriptomic profiling of human bone marrow non-hematopoietic cells | http://www.ncbi.nlm.nih.gov/geo/query/acc.cgi?acc=GSE190965 | NCBI Gene Expression Omnibus, GSE190965 |

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
