## [Editor Report]

The manuscript by Li and coworkers is a landmark characterization of sorted human non-hematopoietic bone marrow cells by scRNA-seq, which predicts their potential lineage relationships and possible interactions with mature and immature hematopoietic cells. Transcriptionally-different stromal cell subsets are identified convincingly, and their lineage relationships, cell-cell interactions and possible specialized functions are predicted from in-silico studies, paving the way for future necessary functional validation studies. This resource significantly adds to the current understanding of human non-hematopoietic bone marrow stromal cells and their hematopoietic regulatory functions.

---

## [Decision Letter]

**Decision letter after peer review:**

Thank you for submitting your article "Single-cell RNA sequencing identifies phenotypically, functionally, and anatomically distinct stromal niche populations in human bone marrow" for consideration by *eLife*. Your article has been reviewed by 2 peer reviewers, and the evaluation has been overseen by a Reviewing Editor and Mone Zaidi as the Senior Editor. The following individuals involved in review of your submission have agreed to reveal their identity: Dirk Strunk (Reviewer #1); Robert A. J. Oostendorp (Reviewer #2).

Collectively, the reviewers agree that your study is an important scRNAseq resource and appreciate the potential interest of your predictions about different cell populations, their lineage relationships, and how they might interact with hematopoietic cells. While functional validations are outside the scope of this inherently descriptive large dataset and would not allow timely publication, the conclusions regarding cell populations and lineage relationships should be toned down and presented as possible suggestions, pending future validations.

Essential revisions:

1) Please provide a glossary containing the abbreviations used to facilitate accessibility.

2) Tone down conclusive statements about predictions, to avoid over-interpretation of the data.

3) Re-analyze or present some of the data to facilitate viewing of the relevant cell-cell interactions or the cell populations identified using candidate markers (specific comments or guidelines are provided by the reviewers in their review).

4) Clarify the conflicting results of adipo/osteogenic commitment inferred from transcriptomic and in vitro differentiation studies.

5) Incorporate textual revisions explained by the reviewers in their detailed comments.

*Reviewer #1 (Recommendations for the authors):*

– Please report the frequency/numbers of sorted CD45low/-CDa-/CD271+ cells and their enriched CXCL subfraction if available.

*Reviewer #2 (Recommendations for the authors):*

– A glossary with all the different abbreviations would be helpful, as some of them are used prior to their definition/description in the text (for instance MSSC).

– Several of the statements in this manuscript are about predictions, speculations, or hypotheses, which appear to be over-interpretions of the data. Thus, these statements should be toned down, unless the authors can provide clear experimental evidence confirming these statements.

– One such statement is in the abstract and mentions specific BM regions "using" CXCL12 or SPP1. This should reformulate, as hard evidence to support this statement is not provided.

---

## [Author Response]

Essential revisions:1) Please provide a glossary containing the abbreviations used to facilitate accessibility.

We thank the reviewer for the suggestion and a list of abbreviations is now provided on page 2.

2) Tone down conclusive statements about predictions, to avoid over-interpretation of the data.

Thank you for this comment. Being aware that we have not provided final experimental proof that validates the potential key drivers of stromal cell fate and the predicted ligand-receptor interactions and functional roles in the current study, we have toned down the corresponding statements as suggested by the reviewer (page 3, lines 11-12; page 18, lines 5-7; page 26, lines 16-21; page 29, lines 18-20; page 31, lines 19-23; page 32, lines 2-5).

3) Re-analyze or present some of the data to facilitate viewing of the relevant cell-cell interactions or the cell populations identified using candidate markers (specific comments or guidelines are provided by the reviewers in their review).

For the cell-cell interaction analysis, reviewer 2 suggested to re-analyze the data to show the specificity of the interactions. However, as shown in our original Figure 5B-C and Figure 5—figure supplement 1B, interactions between MSSC/OC with hematopoietic clusters and the CXCL12- and SPP1-mediated stromal cell type-specific interactions were already presented. Therefore and as outlined in detail in the response to the reviewer, we feel that a re-analysis of the data at this point would not further facilitate viewing of the relevant cellcell interactions.

Reviewer 2 also suggested in the detailed comment that we should describe the possible interactions between non-hematopoietic cells. These interactions were already shown in our original Figure 7A and Figure 5—figure supplement 3B. For the stromal cells, both inter- and intra-stromal interactions were identified to be operative between different stromal subsets as well as within the same stromal cell population (Figure 5—figure supplement 3B). We also analyzed and presented the interaction pattern between endothelial cells and hematopoietic cells (Figure 7A) and a couple of endothelial-specific interactions were detected (SELLSELPLG and ICAM3-aLb2 complex). Therefore, we do not think that additional analysis regarding these interactions is needed.

With regard to the scRNAseq analysis, reviewer 2 suggested in the detailed comments that we should highlight the expression of CD45, CD235a, and CD271 in different analyses and restrict the analyses accordingly. According to this comment, we have modified the figures which are presented in the new Figure R5. However, because the inclusion of these markers in the figures did not provide substantial novel information and because there is a risk of excluding important stromal cells from the analysis, we chose to not make changes to the original figure (Figure 1D, 2B-F and 4A).

Please see also our response to the reviewers’ comments for a detailed discussion of these points.

4) Clarify the conflicting results of adipo/osteogenic commitment inferred from transcriptomic and in vitro differentiation studies.

We thank the editor and the reviewer for this comment. Cluster 5 was initially annotated as “adipo-primed” cluster based on the higher expression of adipogenic differentiation markers as well as a group of stress-related transcription factors (FOS, FOSB, JUNB, EGR1) (Figure 2B-C, Figure 2—figure supplement 1C), some of which had been reported to mark bone marrow adipogenic progenitors^1^. Although at considerably lower levels compared to adipogenic genes, osteogenic genes were also expressed in cluster 5 cells (Figure 2B and D), which indicated a potentially broader differentiation potential of this cluster. As correctly noticed by the reviewers, the initial annotation of this cluster as “adipo-primed progenitors” unfortunately did not reflect this fact correctly. Therefore we have renamed these cells as ‘highly adipocytic gene-expressing progenitors (HAGEPs)” in the revised version of the manuscript. This new name does also more correctly reflect the results from the differentiation studies.

5) Incorporate textual revisions explained by the reviewers in their detailed comments.

We have changed the manuscript according to the reviewers’ comments. Please see our detailed responses below.

*Reviewer #1 (Recommendations for the authors):*
– Please report the frequency/numbers of sorted CD45low/-CDa-/CD271+ cells and their enriched CXCL subfraction if available.

The cell numbers and percentages of sorted CD45^low/-^CD235a^-^ and CD45^low/-^CD235a^-^CD271^+^ cells have been listed in Supplementary File 3 (sheet 1). The numbers and frequency of CXCL12-expressing cells in sorted CD45^low/-^CD235a^-^ and CD45^low/-^CD235a^-^CD271^+^ cells have now bwwn listed in Supplementary File 3 (sheet 4).

Reviewer #2 (Recommendations for the authors):– A glossary with all the different abbreviations would be helpful, as some of them are used prior to their definition/description in the text (for instance MSSC).

We thank the reviewer for the suggestion. A list of abbreviations is now provided on page 2 of the manuscript.

– Several of the statements in this manuscript are about predictions, speculations, or hypotheses, which appear to be over-interpretions of the data. Thus, these statements should be toned down, unless the authors can provide clear experimental evidence confirming these statements.

Thank you for this comment. Being aware that we have not provided experimental proof to validate the role of potential key drivers of stromal cell fate, the predicted ligand receptor interactions and functional roles in the current study, we have toned down the corresponding statements as suggested by the reviewer (page 3, lines 11-12; page 18, lines 57; page 26, lines 16-21; page 29, lines 18-20; page 31, lines 19-23; page 32, lines 2-5).

– One such statement is in the abstract and mentions specific BM regions "using" CXCL12 or SPP1. This should reformulate, as hard evidence to support this statement is not provided.

We have changed the abstract according to the reviewer’s comment (page 3, lines 12-15).

References

Ambrosi TH, Scialdone A, Graja A, et al. Adipocyte Accumulation in the Bone Marrow during Obesity and Aging Impairs Stem Cell-Based Hematopoietic and Bone Regeneration. Cell Stem Cell. 2017;20(6):771-784 e776.

Sutherland DR, Anderson L, Keeney M, Nayar R, Chin-Yee I. The ISHAGE guidelines for CD34+ cell determination by flow cytometry. International Society of Hematotherapy and Graft Engineering. J Hematother. 1996;5(3):213-226.

Boulais PE, Mizoguchi T, Zimmerman S, et al. The Majority of CD45(-) Ter119(-) CD31(-) Bone Marrow Cell Fraction Is of Hematopoietic Origin and Contains Erythroid and Lymphoid Progenitors. Immunity. 2018;49(4):627-639 e626.

Baryawno N, Przybylski D, Kowalczyk MS, et al. A Cellular Taxonomy of the Bone Marrow Stroma in Homeostasis and Leukemia. Cell. 2019;177(7):1915-1932 e1916.

Sotoodehnejadnematalahi F, Staples KJ, Chrysanthou E, Pearson H, Ziegler-Heitbrock L, Burke B. Mechanisms of Hypoxic Up-Regulation of Versican Gene Expression in Macrophages. PLoS One. 2015;10(6):e0125799.

Wang J, Dodd C, Shankowsky HA, Scott PG, Tredget EE, Wound Healing Research G. Deep dermal fibroblasts contribute to hypertrophic scarring. Lab Invest. 2008;88(12):1278-1290. 7. Zhou BO, Yue R, Murphy MM, Peyer JG, Morrison SJ. Leptin-receptor-expressing mesenchymal stromal cells represent the main source of bone formed by adult bone marrow. Cell Stem Cell. 2014;15(2):154-168.

Li H, Ghazanfari R, Zacharaki D, et al. Low/negative expression of PDGFR-α identifies the candidate primary mesenchymal stromal cells in adult human bone marrow. Stem Cell Reports. 2014;3(6):965-974.

Kathiriya JJ, Wang C, Zhou M, et al. Human alveolar type 2 epithelium transdifferentiates into metaplastic KRT5(+) basal cells. Nat Cell Biol. 2022;24(1):10-23.

Wangen JR, Eidenschink Brodersen L, Stolk TT, Wells DA, Loken MR. Assessment of normal erythropoiesis by flow cytometry: important considerations for specimen preparation. Int J Lab Hematol. 2014;36(2):184-196.

Tormin A, Li O, Brune JC, et al. CD146 expression on primary nonhematopoietic bone marrow stem cells is correlated with in situ localization. Blood. 2011;117(19):5067-5077.

Chan CKF, Gulati GS, Sinha R, et al. Identification of the Human Skeletal Stem Cell. Cell. 2018;175(1):43-56 e21.

Pinho S, Lacombe J, Hanoun M, et al. PDGFRalpha and CD51 mark human nestin+ sphereforming mesenchymal stem cells capable of hematopoietic progenitor cell expansion. J Exp Med. 2013;210(7):1351-1367.

Lucas S, Tencerova M, von der Weid B, et al. Guidelines for Biobanking of Bone Marrow Adipose Tissue and Related Cell Types: Report of the Biobanking Working Group of the International Bone Marrow Adiposity Society. Front Endocrinol (Lausanne). 2021;12:744527.

Merrick D, Sakers A, Irgebay Z, et al. Identification of a mesenchymal progenitor cell hierarchy in adipose tissue. Science. 2019;364(6438).

Wolock SL, Krishnan I, Tenen DE, et al. Mapping Distinct Bone Marrow Niche Populations and Their Differentiation Paths. Cell Rep. 2019;28(2):302-311 e305.

de Jong MME, Kellermayer Z, Papazian N, et al. The multiple myeloma microenvironment is defined by an inflammatory stromal cell landscape. Nat Immunol. 2021;22(6):769-780.

Leimkuhler NB, Gleitz HFE, Ronghui L, et al. Heterogeneous bone-marrow stromal progenitors drive myelofibrosis via a druggable alarmin axis. Cell Stem Cell. 2021;28(4):637-652 e638.